



# Flaring efficiencies and NOx emission ratios measured for offshore oil and gas facilities in the North Sea

Jacob T. Shaw[1], Amy Foulds[1], Shona Wilde[2], Patrick Barker[1], Freya Squires[2*], James Lee[2,3], Ruth Purvis[2,3], Ralph Burton[4], Ioana Colfescu[4], Stephen Mobbs[4], Stéphane J.-B. Bauguitte[5], Stuart Young[2], Stefan Schwietzke[6], Grant Allen[1]

[1]Department of Earth and Environmental Sciences, University of Manchester, Oxford Road, Manchester, M13 9PL, UK
[2]Wolfson Atmospheric Chemistry Laboratories, Department of Chemistry, University of York, York, YO10 5DD, UK
[3]National Centre for Atmospheric Science, University of York, York, YO10 5DD, UK
[4]National Centre for Atmospheric Science, School of Earth and Environment, University of Leeds, Leeds, LS2 9JT, UK
[5]Facility for Airborne Atmospheric Measurements FAAM 125, Cranfield University, Cranfield, UK
[6]Environmental Defense Fund, Berlin, Germany
*Now at: British Antarctic Survey, Natural Environment Research Council, Cambridge, CB3 0ET, UK

*Correspondence to*: Grant Allen (grant.allen@manchester.ac.uk)

**Abstract.** Gas flaring is a substantial global source of carbon emissions to atmosphere, and is targeted as a route to mitigating the oil and gas sector carbon footprint, due to the waste of resources involved. However, quantifying carbon emissions from flaring is resource intensive, and no studies have yet assessed flaring emissions for offshore regions. In this work, we present carbon dioxide ($CO_2$), methane ($CH_4$), ethane ($C_2H_6$), and $NO_x$ (nitrogen oxide) data from 58 emission plumes identified as gas flaring, measured during aircraft campaigns over the North Sea (UK and Norwegian) in 2018 and 2019. Median combustion efficiency, the efficiency with which carbon in the flared gas is converted to $CO_2$ in the emission plume, was 98.4% when accounting for $C_2H_6$, or 98.7% when only accounting for $CH_4$. Higher combustion efficiencies were measured in the Norwegian sector of the North Sea compared with the UK sector. Destruction removal efficiencies (DREs), the efficiency with which an individual species is combusted, were 98.5% for $CH_4$, and 97.9% for $C_2H_6$. Median $NO_x$ emission ratios were measured to be 0.003 ppm per ppm $CO_2$ and 0.26 ppm per ppm $CH_4$, and the median $C_2H_6$:$CH_4$ ratio was measured to be 0.11 ppm ppm$^{-1}$. The highest $NO_x$ emission ratios were observed from Floating Production Storage and Offloading (FPSO) vessels, although this could potentially be due to the presence of alternative $NO_x$ sources onboard, such as diesel generators. The measurements in this work were used to estimate total emissions from the North Sea from gas flaring, of 1.4 Tg yr$^{-1}$ $CO_2$, 6.3 Gg yr$^{-1}$ $CH_4$, 1.7 Gg yr$^{-1}$ $C_2H_6$, and 3.9 Gg yr$^{-1}$ $NO_x$.

## 1 Introduction

Gas flaring is a practice widely used at hydrocarbon production sites to dispose of natural gas in situations where the gas is not captured for sale or used locally, and would otherwise be vented directly to atmosphere. Flaring leads to the emission of carbon dioxide ($CO_2$) and short-lived climate forcers such as methane ($CH_4$) and black carbon (BC) (Myhre et al., 2013; Allen





et al., 2016; Fawole et al., 2016; IPCC, 2021). Ideally, all flammable gas would be fully combusted to form $CO_2$ as $CH_4$ is a much more powerful greenhouse gas (Allen et al., 2016). Flaring also results in the emission of combustion by-products, which include carbon monoxide (CO), nitrogen oxides ($NO_x$), and sulphur dioxide ($SO_2$), as well as other components of the unburned

fuel (such as volatile organic compounds, VOCs), which have been known to have adverse health and environmental impacts (Kahforoshan et al., 2008; Anejionu et al., 2015; EPA, 2011). The International Energy Agency (IEA) estimated that $142 \times 10^9$ m$^3$ of natural gas was flared in 2020, resulting in emissions of 265 Tg of $CO_2$ and 8 Tg of $CH_4$ (IEA, 2021). For $CH_4$, this represents roughly 7% of all fossil fuel related emissions, or approximately 2% of total annual anthropogenic emissions (Saunois et al., 2020).  As a large source of greenhouse gas emissions (Olivier et al., 2013), reductions in gas flaring are

required in order to meet emission targets within the Kyoto Protocol's Clean Development Mechanism (United Nations, 1998; Elvidge et al., 2018).

Flaring is typically assumed to be highly efficient. Many inventories assume 98% of flared natural gas is converted to $CO_2$ (EPA, 2011; Allen et al., 2016). However, factors such as the flare volume, flare gas flow rate, or even the strength of ambient winds can affect the efficiency of flares, which can result in incomplete combustion (Johnson and Kostiuk, 2002;

Allen et al., 2016; Jatale et al., 2016). The IEA suggests an alternative globally averaged combustion efficiency of 92%, resulting in emissions of 500 Tg $CO_2$-eq in 2020 (IEA, 2021). Large uncertainties in combustion efficiencies lead to significant uncertainties in total greenhouse gas emissions from flaring (Allen et al., 2016).

There have been minimal real-world studies of flaring combustion efficiencies, with the majority focussed on test facilities and permanent flares that are subject to emission regulations (e.g. Knighton et al., 2012; Torres et al., 2012a, 2012b).

Flaring from oil and natural gas fields is often temporary and in-field sampling is required to gain insight into combustion efficiencies across a wide range of real operating conditions (Ismail and Umukoro, 2012). Caulton et al. (2014) measured the destruction removal efficiency (DRE) of $CH_4$ in 11 flared gas plumes in the Bakken Shale Formation, United States. They found that gas flares were 99.8% efficient at removing $CH_4$, and that wind speeds below 15 m s$^{-1}$ did not have an effect on their efficiency. A similar airborne study of 37 unique flares in the same Bakken region found a skewed log-normal distribution

of flare efficiencies, with median DREs of 97% for both $CH_4$ and ethane ($C_2H_6$) but also some flares with much lower DREs of less than 85% (Gvakharia et al. 2017). The discrepancy in flaring efficiencies measured by these two studies may be due to the targeting of larger flares (which are typically more efficient) by Caulton et al. (2014), but may also have been potentially due to the limited sample sizes. A recent study presented results from a much larger sample of over 300 unique flares measured across three major oil and gas basins in the United States (Bakken Formation, Eagle Ford Shale, and Permian Basin), with

mean observed DREs for $CH_4$ of 95.2% (Plant et al., 2022). The results exhibited a strong skewed-distribution, and, when accounting for the contribution of unlit flares (which vent $CH_4$ directly to atmosphere), the mean effective DRE for $CH_4$ was 91.1% (Plant et al., 2022).

Offshore oil and gas facilities in the North and Norwegian Seas have been the subject of several studies complementary to the work presented here. Foulds et al. (2022) measured $CH_4$ emission fluxes from 21 offshore facilities on

the Norwegian continental shelf, finding mean emissions of 211 tonnes $CH_4$ yr$^{-1}$ (6.7 g $CH_4$ s$^{-1}$) per facility. Wilde et al. (2021a)



measured much larger median $CH_4$ emissions of 120 g $CH_4$ s$^{-1}$ (range: 20-360 g $CH_4$ s$^{-1}$) from four facilities in the North Sea. Riddick et al. (2019) measured $CH_4$ emissions using a shipborne platform, reporting median emissions of 6.8 g $CH_4$ s$^{-1}$ (214 tonnes $CH_4$ yr$^{-1}$) across eight facilities, in exceptional agreement with Foulds et al. (2022). In the southern North Sea, Pühl et al. (*in prep*) measured median emissions of 10 g $CH_4$ s$^{-1}$ from a sample of UK and Dutch oil and gas platforms. However, Pühl

et al. (*in prep*) also measured emissions of 350 g $CH_4$ s$^{-1}$ from a single platform, similar in magnitude to the largest emitters measured by Wilde et al. (2021a). The discrepancies between these emission flux estimates, which are often based on 'snapshot' studies conducted over limited timeframes, may be due to capturing different events, measuring at different lifetime phases of production, or small sample sizes. Shipborne-based measurements may also fail to capture flared emissions, as these are typically warmer than ambient air and would therefore be expected to rise in the atmosphere. The carbon isotopic signature

of methane emitted from oil and gas facilities is useful for source identification, and has been measured to be -53‰ in the North Sea (Cain et al., 2017; France et al. 2021). Emissions of volatile organic compounds (VOCs) from oil and gas facilities have also been measured in the North Sea, with ratios in enhancements of $C_2H_6$ to $CH_4$ ($\Delta C_2H_6$:$\Delta CH_4$) measured to be between 0.03 and 0.18 ppm ppm$^{-1}$ (Wilde et al., 2021a; 2021b; Pühl et al., *in review*).

       The volume of gas flared in the UK North Sea was reported to have fallen by 19% in 2021 (OGA, 2021). Despite

this, 740 million cubic metres ($7.4 \times 10^8$ m$^3$) of natural gas were still flared (OGA, 2021), equivalent to 0.5% of gas flared globally. The UK was 23$^{rd}$ in the list of countries with the greatest total flaring volumes for 2020 (World Bank, 2021), with the top seven countries accounting for 65% of all flaring. The Zero Routine Flaring initiative, launched in 2015, aims to end routine gas flaring no later than 2030 and hence emissions from flaring must be monitored. Monitoring current flaring emissions from the oil and gas sector is therefore essential to robustly assess any future changes or reductions to flaring activity.

In this work, we present combustion efficiencies, destruction removal efficiencies (DREs), and NO$_x$ emission ratios calculated for a sample of flared gas plumes measured across two aircraft campaigns in the North and Norwegian Seas.



## 2 Methods

### 2.1 Atmospheric research aircraft

All flight measurements analysed in this work were made using the UK's Facility for Airborne Atmospheric Measurement (FAAM) BAe-146 atmospheric research aircraft. A description of the full aircraft scientific payload can be found in Palmer et al. (2018). Here, we summarise the instrumentation relevant to this study.

Meteorological and thermodynamic parameters were measured using the core instrument suite onboard the FAAM aircraft. A Rosemount 102 Total Air Temperature probe measured air temperature with an estimated uncertainty of ±0.1 K.
Static pressure was measured using a series of pitot tubes (uncertainty ±0.5 hPa) and a nose-mounted five-port turbulence probe measured three-dimensional wind components (uncertainty ±0.5 m s$^{-1}$).

Dry mole fractions of $CO_2$ and $CH_4$ were measured using a cavity enhanced absorption spectrometer (Fast Greenhouse Gas Analyzer (FGGA); Los Gatos Research Inc., USA), sampling air through a window-mounted rear-facing chemistry inlet. A full description of the FGGA for measurements onboard the FAAM aircraft was reported by O'Shea et al. (2013), with a
modified instrumental setup (used after January 2019) described by Shaw et al. (2022). Raw $CO_2$ and $CH_4$ mole fraction data were corrected for small effects associated with water vapour dilution and spectroscopic error. Calibration was performed approximately hourly during flights, using two reference calibration gas cylinders (encapsulating a representative range of background and in-plume mole fractions) traceable to the WMO-X2007 scale for $CO_2$ (Tans et al., 2009) and the WMO-X2004A scale for $CH_4$ (Dlugokencky et al., 2005). A target reference gas cylinder was also sampled hourly to quantify small
sources of instrumental drift and non-linearity, and to define measurement error. For a full description of data correction, calibration and validation, refer to O'Shea et al. (2013) and Pitt et al. (2019). $CH_4$ and $CO_2$ data were measured at 1 Hz for flights conducted in 2018, and at 10 Hz for flights conducted in 2019 (Foulds et al., 2022; Shaw et al., 2022). 10 Hz measurements were time-averaged onto a 1 Hz grid for consistency between datasets. The representative one standard deviation (1σ) measurement uncertainties were ±2.86 ppb $CH_4$ and ±0.46 ppm $CO_2$ at a sampling rate of 1 Hz, and ±3.23 ppb $CH_4$ and
±0.72 ppm $CO_2$, at 10 Hz.

Ethane ($C_2H_6$) mole fractions were measured using a tunable infrared laser direct absorption spectrometer (TILDAS, Aerodyne Research Inc.), operating at 1 Hz in the mid-infrared region (λ = 3.3 μm). Raw $C_2H_6$ mole fraction data were corrected for spectroscopic effects associated with water vapour using the method described by Pitt et al. (2016). Calibration was performed using two gas standards (encapsulating a range of mole fractions) certified by the Swiss Federal Laboratories
for Materials Science and Technology (EMPA). The TILDAS instrument has a reported precision of ±50 ppt over a 10 s averaging period. Two levels of data quality were provided for the $C_2H_6$ dataset. The "high quality" data included data that were calibrated at a stable altitude to account for systematic biases from optical effects (see Pitt et al., 2016). The "reduced quality" data included regular linear calibration (at ~45 minute intervals) but included data where calibration was not possible at a stable altitude. However, as we use enhanced $C_2H_6$ mole fractions (background subtracted) in this work, the systematic
altitude-dependent biases were effectively removed, and the "reduced quality" $C_2H_6$ data was considered acceptable.





Nitrogen monoxide (NO) and nitrogen dioxide ($NO_2$) were measured using a custom-built chemiluminescence instrument (Air Quality Design Inc.; see Graham et al., 2020 and Lee et al., 2009 for detail). $NO_2$ was measured on a secondary channel following photolytic conversion to NO using a blue light converter (395 nm), and subsequent detection via chemiluminescence. In-flight calibrations were performed frequently using a small flow of NO calibration gas (5 ppm NO in

$N_2$). Estimated accuracies were ±4% for NO and ±5% for $NO_2$, with precisions of 31 and 45 pptv for NO and $NO_2$ respectively at 1 Hz. NO and $NO_2$ mole fractions below the instrument detection limit of 30 pptv were removed.

All instrumentation on board the FAAM aircraft were synchronised with respect to time prior to each flight. However, instrument-specific temporal drift led to small temporal discrepancies between instruments during some flights. In cases where identified plumes were misaligned in time, data were manually corrected to align the peaks where possible.

Data availability from some instruments for some flights was limited (see Table A1). The $NO_x$ instrument suffered from large data gaps in three AEOG flights. This may have been because local $NO_x$ background mole fractions were below the instrument limit-of-detection (30 pptv). However, data availability within plumes was also affected for these, and other, flights.

## 2.2 Flight sampling and study areas

This work used data collected as part of two field measurement campaigns: the Assessing Atmospheric Emissions from the Oil and Gas Industry (AEOG) programme, and the Methane Observations and Yearly Assessments (MOYA) project. The AEOG flights targeted two key production regions on the UK continental shelf (UKCS). A total of 14 flights over the North Sea in the northern UK and West Shetland region were conducted in April 2018, September 2018, or March 2019. The MOYA campaign involved three flights in July and August 2019, surveying two regions on the Norwegian continental shelf (one in

the North Sea, and one in the Norwegian Sea). Figure 1 shows flight tracks for the AEOG and MOYA campaigns, as well as the offshore hydrocarbon fields and corresponding field types.





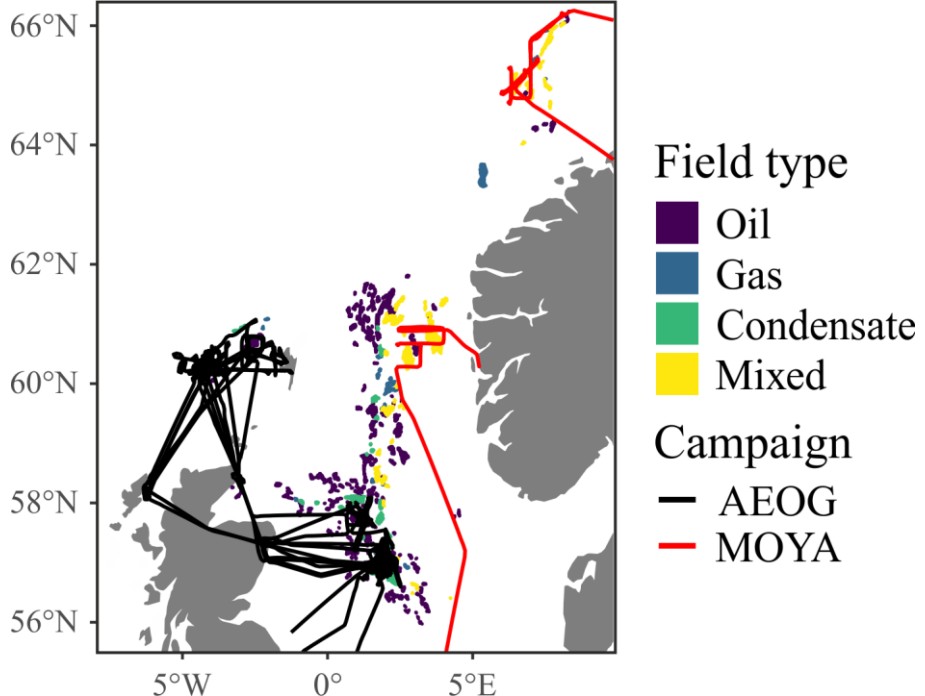

**Figure 1. AEOG (black) and MOYA (red) flight paths in the North and Norwegian Seas. Coloured data points indicate the locations**
**of different hydrocarbon field types (see Wilde et al., 2021b or Foulds et al., 2022 for detail). Note that the northern-most flight (~65°**
**N) took place over the Norwegian Sea and not the North Sea. However, for the purposes of simplicity here, all sample regions are**
**referred to as the North Sea.**

### 2.3 Identification of flared emissions and flaring efficiency calculations

Gas flaring could not be confirmed visually during the flight campaigns due to distance to targeted facilities. In the absence of visual flare confirmation, plumes associated with gas flaring were identified by correlated enhancements in the expected gas-phase components of flared hydrocarbon gas (i.e. $CO_2$, $CH_4$, $C_2H_6$, and $NO_x$) above their respective background mole fractions. Plumes which did not contain correlated enhancements of all four of these components were discarded. For example, plumes containing enhancements in only $CO_2$, $CH_4$ and $C_2H_6$, and which therefore lacked enhancements in $NO_x$, were discarded as they were assumed result from gas venting without flaring. Similarly, plumes containing only enhancements in $CO_2$ and $NO_x$, and therefore lacking enhancements in either of $CH_4$ or $C_2H_6$, were assumed to result from emissions from power generation, such as diesel generators. Unfortunately, this approach does not preclude the possibility of including emissions from multiple mixed sources of $CO_2$, $CH_4$, $C_2H_6$, or $NO_x$, such as co-located venting and power generation emissions.

Representative median-average background $CO_2$, $CH_4$, $C_2H_6$ and $NO_x$ mole fractions were determined for each plume using the 50 neighbouring 1 Hz measurements to either side of the plume. Plumes for which this was not possible due to missing background data for one or more components (i.e. fewer than 10 background data points) were discarded. Plumes were additionally discarded if one or more components lacked sufficient data within the plume (i.e. fewer than three data points). The $NO_x$ data generally suffered from data unavailability (see Table A1), with large proportions of missing 1 Hz data.





During background measurement, missing data could be attributed largely to $NO_x$ mixing ratios below the instrument limit-of-detection (30 pptv) but missing data within plumes was also common. If enough data were present, missing $NO_x$ data were

interpolated using normalised values of the $CO_2$ and $CH_4$ plume data. Figure 2 shows an example in which three missing data points within a single plume were interpolated and reconstructed using the mean-average normalised $CO_2$ and $CH_4$ data. Using this method relies on the assumption that each gas has an identical plume morphology, which may not always be the case if there are multiple co-located sources upwind (France et al., 2021). However, Fig. 2 clearly demonstrates that all four gas components showed consistent plume morphologies in this example. Finally, plumes were discarded if the maximum within-

plume enhancement was within two standard deviations (2σ) of the local background mole fraction.

Background mole fractions were subtracted from within-plume mole fractions to calculate enhancements. The resultant plume enhancements were then integrated (with respect to time) to determine the amount of each component within the emission plume. Integrating the data, rather than performing linear regression of co-located components, allows for slight temporal discrepancies in measured plumes to be ignored. Temporal discrepancies which lead to misaligned plumes could

affect linear correlations between plume components.

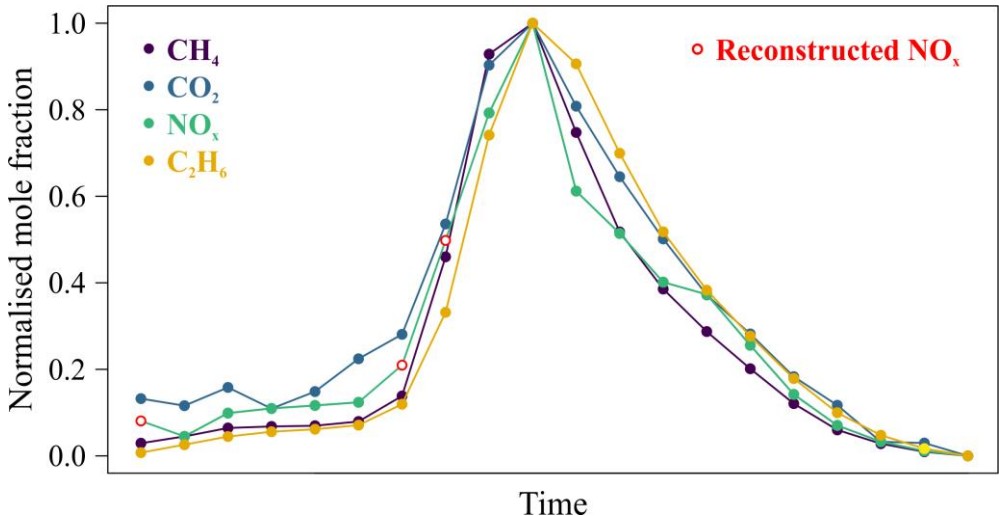

**Figure 2.** Normalised mole fraction (enhancements above background) for a plume containing $CO_2$, $CH_4$, $C_2H_6$, and $NO_x$. Three $NO_x$ data points were missing and were interpolated and reconstructed using the mean of normalised mole fractions of $CO_2$ and $CH_4$ (red

data points).

### 2.3.1 Combustion efficiency calculations

Combustion efficiency ($\eta$) can be defined in multiple ways, but is usually reported as the efficiency with which the gas flare converts hydrocarbons in the fuel gas into carbon dioxide (Equation 1; Corbin and Johnson, 2014).

$$\eta[\%] = \frac{carbon\ in\ CO_2\ in\ flared\ gas}{carbon\ in\ hydrocarbon\ fuel\ gas} \times 100 \qquad \text{(Eq. 1)}$$





However, in many cases, the amount of carbon in the industrial fuel gas is unknown. Fuel composition can vary widely between production regions and within fields, as well as over the course of production. In cases where fuel composition is not known, combustion efficiencies have previously been approximated using the relationship between enhancements of $CO_2$ and $CH_4$ measured within the flare plume (Equation 2; Nara et al., 2014).

$$\eta[\%] = \frac{\Delta CO_2}{\Delta CO_2 + \Delta CH_4} \times 100 \qquad \text{(Eq. 2)}$$

$\Delta CO_2$ and $\Delta CH_4$ respectively refer to the enhancement of within-plume $CO_2$ and $CH_4$ above the local background mole fractions (see Section 2.3). The method presented in Equation 2 assumes that all of the $CO_2$ produced during gas flaring is due to combustion of $CH_4$ i.e. no other hydrocarbons were combusted (the fuel gas is 100% $CH_4$), and that $CO_2$ was not initially present in the fuel gas. This can lead to a slight overestimation of combustion efficiency, if other hydrocarbons were present in the fuel gas and combusted.

As $C_2H_6$ mole fractions were also measured onboard the FAAM aircraft, additional combustion efficiencies were calculated which account for the $C_2H_6$ enhancement within the plumes (Equation 3). $C_2H_6$ oxidises to form two molar equivalents of $CO_2$, and is therefore accounted for twice in Eq. 3.

$$\eta[\%] = \frac{\Delta CO_2}{\Delta CO_2 + \Delta CH_4 + (2 \times \Delta C_2H_6)} \times 100 \qquad \text{(Eq. 3)}$$

Where $\Delta C_2H_6$ refers to the enhancement of within-plume $C_2H_6$ above the local background mole fraction. It should be noted

that combustion efficiencies calculated with Eq. 3 will still overestimate the true combustion efficiency by some amount. Although $CH_4$ and $C_2H_6$ typically dominate the fuel gas composition, other hydrocarbons are likely to be present (albeit, in small amounts) and cannot be accounted for here. However, this approach provides the best possible approximation in the absence of suitable instrumentation capable of resolving larger hydrocarbons at 1 Hz.

**2.3.2 Destruction removal efficiency calculations**

Destruction removal efficiency (DRE) is a measure of the efficiency with which a particular fuel gas component is oxidised within the flare (Equation 4; Caulton et al., 2014; Corbin and Johnson, 2014).

$$DRE_i[\%] = \left(1 - \frac{\Delta x_i}{(X_i \times \Delta CO_2) + \Delta x_i}\right) \times 100 \qquad \text{(Eq. 4)}$$

Where $x_i$ refers to any component of the fuel gas, $\Delta x_i$ the enhancement above background of that component within the plume, and $X_i$ is the fractional composition of $x_i$ in the fuel gas. Equation 4 was used to calculate DREs for $CH_4$ and $C_2H_6$.

Fuel gas composition values for various platforms were taken from privately communicated fuel composition data sourced via the Department for Business, Energy, and Industrial Strategy (BEIS). Where gas flare plumes could be satisfactorily attributed to single platforms (or groups of platforms), specific fuel composition values were used for $X_i$. In the absence of data for identified platforms, or where plumes could not be satisfactorily associated with specific platforms, the median fuel composition of all available data was used. The median fuel composition for $CH_4$ was 0.845, and for $C_2H_6$ was





0.085. These fuel compositional values are consistent with those used in other work (e.g. Schwietzke et al., 2014; Sherwood et al., 2017).

### 2.3.3 Emission ratio calculations

$NO_x$ and $C_2H_6$ emission ratios (ERs) were calculated using $CO_2$ and $CH_4$ as the reference gas component.

$$ER_{NO_x} = \frac{\Delta NO_x}{\Delta CO_2} = \frac{NO_{x,plume} - NO_{x,background}}{CO_{2,plume} - CO_{2,background}} \tag{Eq. 5}$$

$$ER_{C_2H_6} = \frac{\Delta C_2H_6}{\Delta CH_4} = \frac{C_2H_{6,plume} - C_2H_{6,background}}{CH_{4,plume} - CH_{4,background}} \tag{Eq. 6}$$

ERs calculated in this way are also referred to as normalised excess mixing ratios (NEMRs), and assume that no chemical processing has occurred within the plume that could change the composition (Yokelson et al., 2013; Barker et al., 2020). This assumption is suitable for the components analysed here, as plumes were typically measured less than 10 km downwind of the source. The atmospheric lifetimes of $CH_4$ (~9 years; Turner et al., 2017), and $C_2H_6$ (~2 months; Hodnebrog et al., 2018) ensure

minimal chemical processing, and $NO_x$ is a conserved quantity unaffected by the conversion of NO to $NO_2$ between emission and measurement.

### 2.4 Gas flaring emission inventories

Many emission inventories group emissions from the oil and gas sector into a single category, representing intentional venting, flaring, and leakage. The two emission inventories used here provide separate categories for flaring emissions.

230            The Global Fuel Exploitation Inventory (GFEI) is a globally gridded inventory of $CH_4$ emissions from oil, gas, and coal exploitation, available at $0.1° \times 0.1°$ for 2019 (Scarpelli et al., 2020). The GFEI provides gridded emissions from different subsectors (e.g. exploration, production, transport, transmission, and refining), and from specific processes such as venting and flaring, based on country reports submitted in accordance with the United Nations Framework Convention on Climate Change (UNFCCC). $CH_4$ emissions from flaring during gas production, gas processing, and oil production were examined

here. In the GFEI, $CH_4$ emissions from flaring during oil exploration, gas exploration, and oil refining are grouped together with emissions from leakage and venting, and hence these emissions were not analysed. Comparisons between the GFEI and $CH_4$ emission fluxes measured in the North Sea have already been made by Foulds et al. (2021) and Pühl et al. (in review).

            The anthropogenic emission dataset Evaluating the Climate and Air Quality Impacts of Short-lived Pollutants (ECLIPSE) v5 provides global $CH_4$ and $NO_x$ emissions (amongst other pollutants) for flaring as a separate sub-sector, at $0.5°$

$\times 0.5°$ resolution for 2020 (Stohl et al., 2015). The ECLIPSE emission dataset was created using the Greenhouse gas-Air Pollution Interactions and Synergies (GAINS) model and international and national activity data for energy usage, industrial production, and agricultural activities.





# 3 Results and discussion

Fifty-eight plumes from a maximum of 30 individual facilities were identified as containing emissions from gas flaring based
on the criteria described in Section 2.3 (see Table A2 for numbers of excluded plumes). As some plumes from the same facility
were sampled multiples times, there are two conceivable approaches to determining plume statistics. Firstly, measurements
for plumes considered to originate from the same source could be combined, assuming that the combustion efficiency and
emission ratios are constant. This would allow for uncertainty estimation, using the variability in the measured values.
However, this may not be trivial as changing conditions (in e.g. wind direction) could mean that plumes do not always appear
in the same location and therefore cannot always be positively attributed to the exact same source (in the absence of complex
and time-consuming dispersion modelling). A second approach involves treating each intercepted plume as unique, by
assuming that flaring conditions vary over time and that separate plume intercepts represent distinct measurements of
instantaneous emissions. In this work, it was noted that plumes considered to have the same source origin (via approximate
wind direction) had similar $\Delta C_2H_6{:}\Delta CH_4$ emission ratios but that combustion efficiency varied with wind speed (see Appendix
B). Hence, we have opted to treat the 58 identified plumes as individual and unique events. The following sections therefore
present combustion efficiency, destruction removal efficiencies (for $CH_4$ and $C_2H_6$) and emission ratio results for the 58
identified plumes.

Figure 3 illustrates the relative abundance of gaseous components in the 58 sampled flared plumes. As expected, $CO_2$
was the largest component by at least an order of magnitude. The range in $CH_4$, $C_2H_6$, and $NO_x$ spanned greater than two orders
of magnitude. This could imply the measurement of emissions from flares of different operational characteristics and fuel gas
volumes.



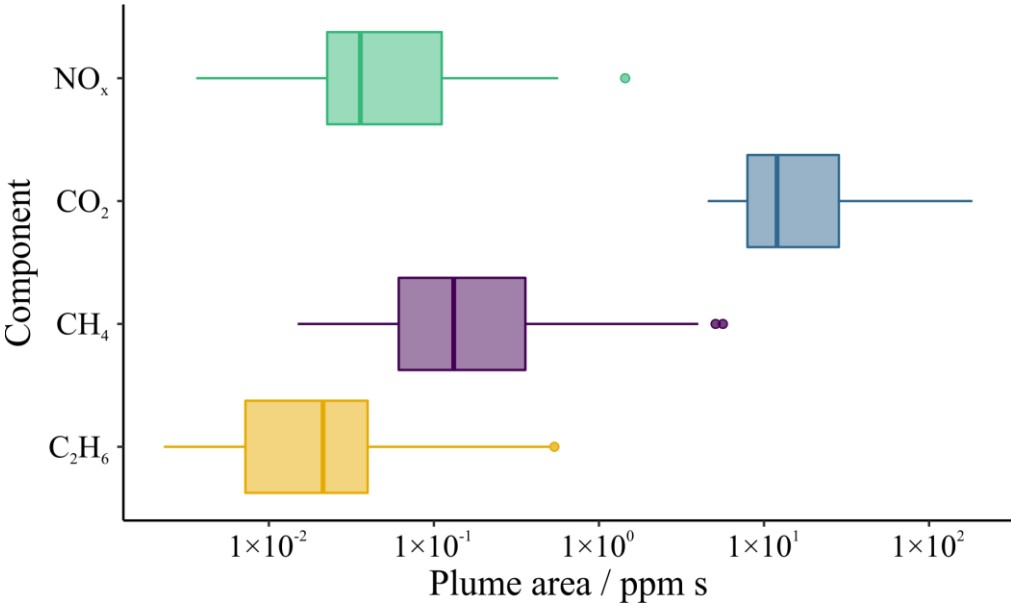

**Figure 3. Box and whisker distributions of integrated plume areas (in ppm s) above background for NOx, CO₂, CH₄, and C₂H₆ across the 58 identified flaring plumes. Box edges correspond to the first and third quartile (i.e. the 25th and 75th percentile) with the thicker, central line denoting the sample median (i.e. 50th percentile). The upper whisker extends to the greatest value no more than 1.5 multiples of the interquartile range (IQR) from the 75th percentile value. The lower whisker extends to the smallest value no less than 1.5 multiples of the IQR from the 25th percentile. Data beyond the extents of the whiskers were considered outlying points, and were plotted individually (as circles). Note that the x-axis has a logarithmic scale.**





### 3.1 Combustion efficiency

Figure 4a shows the distribution of combustion efficiencies calculated without $C_2H_6$ (Eq. 2; Nara et al., 2014) and with $C_2H_6$ included (Eq. 3). Combustion efficiencies were marginally greater when $C_2H_6$ was not included in the calculation. However,
even when including $C_2H_6$ in the calculation, efficiencies were high, with some plumes approaching 100% efficiency, and all efficiencies greater than 94%. The median combustion efficiency across all sampled plumes without $C_2H_6$ included was 98.7% (mean = 98.3% ± 1.4%, 1σ), and the median efficiency with $C_2H_6$ included was 98.4% (mean = 97.9% ± 1.7%, 1σ) (see also Fig. C1). These values are exceptionally close to the 98% combustion efficiency assumed by many emission inventories. However, Fig. 4a shows a strongly skewed distribution, indicating that assumptions of 98% combustion efficiency is likely to
be an overestimate in some cases. A summary of all results can be found in Table 1.

Figure 4b shows the linear relationship between combustion efficiencies calculated with and without $C_2H_6$. The linear relationship was estimated using reduced major axis regression. Combustion efficiencies calculated including $C_2H_6$ (Eq. 3) were marginally smaller than those calculated without $C_2H_6$ (Eq. 2). This relationship provides an approximation for estimating combustion efficiencies accounting for $C_2H_6$ in the absence of direct $C_2H_6$ observations. The $R^2$ value for the linear regression
was 0.996, indicating a high degree of model fit.

There was a small difference in combustion efficiencies (calculated including $C_2H_6$) measured during the AEOG and MOYA campaigns. The median combustion efficiency measured during AEOG (n = 46 plumes) was 97.6% (mean = 97.5% ± 1.6%, 1σ) whilst the median combustion efficiency measured during MOYA (n = 12) was 99.6% (mean = 99.4% ± 0.6%, 1σ). We cannot provide a conclusive explanation for this small difference in combustion efficiencies between the two campaigns
but propose two explanations. AEOG sampled primarily UK-based platforms whilst MOYA sampled Norwegian platforms. It may therefore be possible that differences in facility type, age, or operational practices in the two regions were responsible for the observed distinction in combustion efficiency. Alternatively, the measurements could be explained by differences in emissions from different hydrocarbon field types (see Fig. 1) with different gas compositions. Wilde et al. (2021b) measured different VOC compositions in emissions from different field types in the North Sea region, and this may align with differences
in the combustion efficiency observed here. However, Plant et al. (2022) found no correlation between combustion efficiency and factors such as well age, or gas-to-oil ratio, for onshore facilities in the USA.







**Figure 4. a)** Histogram distribution of combustion efficiencies ($\eta$) calculated with $C_2H_6$ (green; Eq. 3) and without $C_2H_6$ (orange; Eq. 2). **b)** Linear relationship between combustion efficiencies calculated with $C_2H_6$ and without $C_2H_6$. The solid black line shows the linear reduced major axis regression, with $R^2 = 0.996$. The dashed black line shows a 1:1 ratio.



**Table 1. Summary of combustion efficiency, destruction removal efficiency (DRE) and emission ratio results.**

| Measurement (n = 58) | Median | Mean (± 1σ) |
|---|---|---|
| Combustion efficiency (without $C_2H_6$) | 98.7% | 98.3% (± 1.4%) |
| Combustion efficiency (with $C_2H_6$) | 98.4% | 97.9% (± 1.7%) |
| DRE $CH_4$ | 98.5% | 97.9% (± 1.7%) |
| DRE $C_2H_6$ | 97.9% | 97.6% (± 1.7%) |
| $\Delta NO_x{:}\Delta CO_2$ | 0.003 ppm ppm$^{-1}$<br>0.002 g g$^{-1}$* | 0.004 (± 0.004) ppm ppm$^{-1}$<br>0.003 (± 0.003) g g$^{-1}$* |
| $\Delta NO_x{:}\Delta CH_4$ | 0.26 ppm ppm$^{-1}$<br>0.63 g g$^{-1}$* | 0.48 (± 0.65) ppm ppm$^{-1}$<br>1.14 (± 1.54) g g$^{-1}$* |
| $\Delta C_2H_6{:}\Delta CH_4$ | 0.11 ppm ppm$^{-1}$<br>0.20 g g$^{-1}$ | 0.13 (± 0.06) ppm ppm$^{-1}$<br>0.24 (± 0.11) g g$^{-1}$ |

*Assumes a $NO_x$ molar mass of 38.01 g mol$^{-1}$, equivalent to 50% NO and 50% $NO_2$.


Whilst combustion efficiency is expected to decrease with increasing wind speed (Jatale et al., 2016), recent studies have found little-to-no impact on flaring efficiency at wind speeds of up to 15 m s$^{-1}$ (Caulton et al., 2014; Plant et al., 2022). Figure 5 shows an extremely weak but positive correlation ($p = 0.04$; $R^2 = 0.08$) between combustion efficiency and wind speed across

the 58 identified plumes, although there was much scatter in the data. The observed trend was likely skewed by the greater number of plumes sampled under wind speeds of approximately 15 m s$^{-1}$, several of which were measured during the MOYA campaign. Plumes sampled during the MOYA campaign had typically higher combustion efficiencies and therefore may be influencing the observed trend. The only plume measured in wind speeds of approximately 20 m s$^{-1}$ (19.6 m s$^{-1}$) showed a lower combustion efficiency (~95.0%) relative to many of those measured at wind speeds of 15 m s$^{-1}$. Unfortunately, this was

an isolated measurement and a larger sample size of plumes sampled under higher wind speeds (> 15 m s$^{-1}$) would be required to draw meaningful conclusions on combustion efficiencies at such wind speeds. Our results were therefore in agreement with the conclusions of both Caulton et al. (2014) and Plant et al. (2022), which both showed no statistical relationship between combustion efficiency and wind speed.






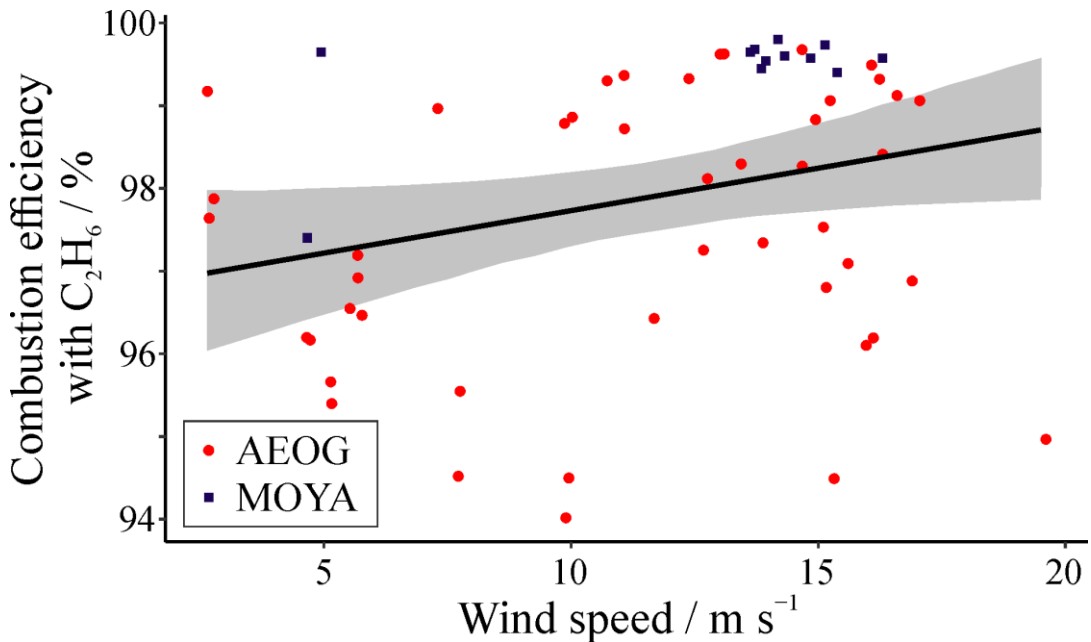

**Figure 5. Correlation between combustion efficiency (calculated including $C_2H_6$; Eq. 3) and wind speed (m s$^{-1}$). The wind speed for each plume was calculated as the mean of 1 Hz wind speeds measured during, and both 50 seconds before and after the plume. The black line shows an ordinary least squares linear regression of the data ($p = 0.04$, $R^2 = 0.08$), with the 95% confidence interval shown in grey.**






### 3.2 Destruction removal efficiencies (DREs)

Figure 6a shows the distribution of DREs calculated for both $CH_4$ and $C_2H_6$ using Eq. 4 and fuel composition data provided by BEIS. The efficiency of $CH_4$ destruction was marginally greater than that for $C_2H_6$, with median values of 98.5% (mean = 97.9% ± 1.7%, 1σ) and 97.9% (mean = 97.6% ± 1.7%, 1σ) for $CH_4$ and $C_2H_6$ respectively (Table 1; see also Fig. C2). Gvakharia et al. (2017) reported marginally lower median DRE values of 97.1% (± 0.4%) for $CH_4$, and of 97.3% (± 0.3%) for $C_2H_6$, from 37 flare plumes in the Bakken formation, United States. Plant et al. (2022) reported mean DRE values for $CH_4$ of 97.3%, 96.5%, and 91.7% from the Eagle Ford, Bakken, and Permian basins (United States) respectively. These results are in excellent agreement with our own. Figure 6b shows the relationship between DREs for the two fuel components, with a strong correlation between the two, even for DREs calculated for plumes from platforms for which flare gas composition was not available (see Section 2.3.2).

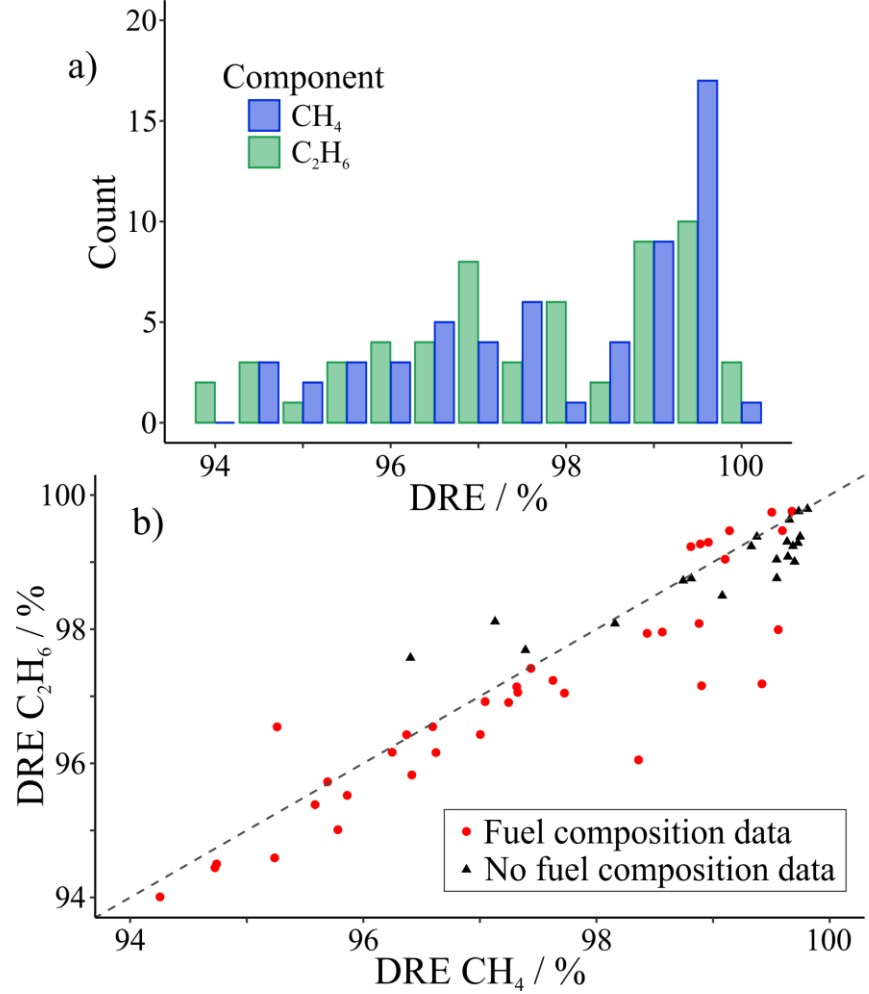



**Figure 6. a) Histogram distribution of destruction removal efficiencies (DREs) calculated for CH$_4$ (blue) and for C$_2$H$_6$ (green). b) Comparison of DREs for CH$_4$ and C$_2$H$_6$. The black dashed line shows a 1:1 ratio. The median fuel composition (CH$_4$ = 0.845, C$_2$H$_6$ = 0.085) was used for plumes emitted from platforms for which no fuel composition data were available (black triangles).**

### 3.3 Emission ratios

Figure 7 shows the distribution of NO$_x$ emission ratios calculated using both CO$_2$ and CH$_4$ as reference gases ($\Delta$NO$_x$:$\Delta$CO$_2$ and $\Delta$NO$_x$:$\Delta$CH$_4$ respectively). Mean NO$_x$ emission ratios were 0.004 ± 0.004 (1σ; median = 0.003) ppm ppm$^{-1}$ when using CO$_2$ as the reference gas, and 0.48 ± 0.65 (1σ; median = 0.26) ppm ppm$^{-1}$ when using CH$_4$ as the reference gas (Table 1). There was substantial variability in the amount of NO$_x$ produced relative to both CO$_2$ and CH$_4$, as indicated by the large standard deviations about the mean ratios and the skewed long-tail distributions in both Fig. 7a and 7b. This may be a consequence of the inclusion of mixed emission sources within our dataset; it is difficult to distinguish between plumes containing pure flaring emissions, and those potentially containing mixed emissions from co-located sources.

Four of the five greatest $\Delta$NO$_x$:$\Delta$CH$_4$ ratios (>1.1 ppm ppm$^{-1}$) were measured over deep-water oilfields west of the Shetland Isles, where oil production is typically performed by Floating Production Storage and Offloading (FPSO) vessels. An additional high $\Delta$NO$_x$:$\Delta$CH$_4$ ratio (of 1.5 ppm ppm$^{-1}$) was measured in a shallow water field, east of Scotland, also operated by an FPSO. FPSO vessels have been reported to contribute to 21% of all offshore flaring volume (Charles and Davis, 2021), and the high $\Delta$NO$_x$:$\Delta$CH$_4$ ratios measured in the vicinity of their operation here could indicate a difference in operational practice (e.g. diesel generators onboard FPSO vessels contributing to NO$_x$ emissions) compared with fixed platforms. The same five FPSO plumes also had the five greatest $\Delta$NO$_x$:$\Delta$CO$_2$ ratios.

Typically, NO$_x$ emissions from flares are estimated using emission factors and activity rates, and often use flare heat as a proxy for NO$_x$ emission rates. Torres et al. (2012c) reported a mean NO$_x$:CO$_2$ ratio of 0.00020 (± 0.00014) ppb ppb$^{-1}$ from 24 test flares operated under a range of conditions (fuel gas composition, fuel gas flow, lower heating value, and steam or air assisted flow). In comparison, the smallest $\Delta$NO$_x$:$\Delta$CO$_2$ ratio measured in this study was 0.0005 ppb ppb$^{-1}$.





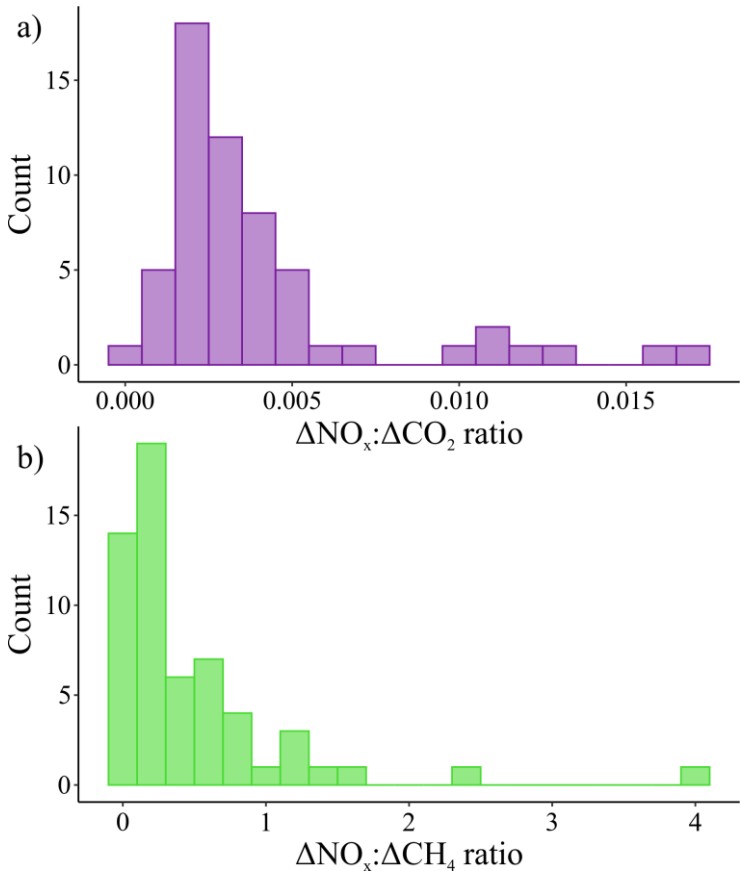

**Figure 7. Histogram distribution of NOₓ emission ratios (ppm ppm⁻¹) a) calculated using CO₂ as the reference gas; and b) calculated using CH₄ as the reference gas.**


Figure 8a shows the relationships between combustion efficiency (calculated with $C_2H_6$) and $\Delta NO_x{:}\Delta CH_4$. Higher combustion efficiencies were typically associated with higher relative amounts of $NO_x$, consistent with higher temperature flaring. Figure 8a appears to show an exponential relationship between combustion efficiency and $\Delta NO_x{:}\Delta CH_4$, but a linear regression is also shown for comparison ($p = 9.3 \times 10^{-5}$; $R^2 = 0.24$). $NO_x$ only appeared to be produced in substantial amounts (relative to $CH_4$)

at combustion efficiencies greater than ~96%, with a general increase in $NO_x$ ratios with increasing combustion efficiency beyond this point. However, plumes measured during the MOYA campaign appeared to have reduced $NO_x$ ratios relative to many of those measured in AEOG, despite having greater combustion efficiencies, implying possible differences in flare operation. Torres et al. (2012c) found a similar result, with minimal $NO_x$ produced below a combustion efficiency threshold of roughly 80%, above which $NO_x$ production increased roughly linearly. Wind speed appeared to have very little influence

on $NO_x$ emission ratios ($p = 0.2$; $R^2 = 0.03$) (Fig. 8b).



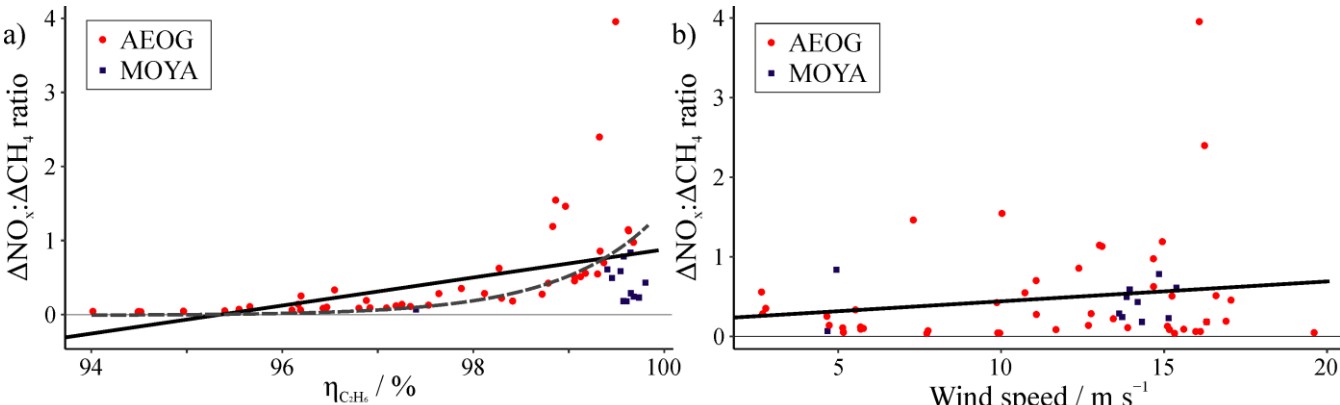

**Figure 8. Correlation between measured $\Delta NO_x:\Delta CH_4$ ratio and a) combustion efficiency calculated with $C_2H_6$ ($\eta_{C_2H_6}$), and; b) wind speed. Solid black lines show ordinary least squares linear regressions with $p = 9.3 \times 10^{-5}$ and 0.2, and $R^2 = 0.24$ and 0.03, for the relationship with combustion efficiency and wind speed respectively. Dashed black line shows the exponential relationship ($y = e^x$) between $\Delta NO_x:\Delta CH_4$ and combustion efficiency, for comparison.**

The mean $\Delta C_2H_6:\Delta CH_4$ ratio across all gas flaring plumes was $0.13 \pm 0.06$ ($1\sigma$) ppm ppm$^{-1}$, with ratios ranging between 0.04 and 0.33 (median = 0.11) ppm ppm$^{-1}$ (Table 1; Fig. C3). These results were in excellent agreement with measurements reported by Wilde et al. (2021b), in which $\Delta C_2H_6:\Delta CH_4$ ratios ranged between 0.03 and 0.18 ppm ppm$^{-1}$. Ratios of between 0.03 and 0.08 ppm ppm$^{-1}$ were also measured for oil and gas emissions in the southern North Sea (Pühl et al., in review). $\Delta C_2H_6:\Delta CH_4$ ratios greater than 0.1 ppm ppm$^{-1}$ are typically associated with emissions from oil wells, whilst ratios below 0.1 ppm ppm$^{-1}$ are usually associated with emissions from gas wells (Xiao et al., 2008; Wilde et al., 2021a).





### 3.4 Emission inventories

The ECLIPSE inventory contains flaring emission products for both $CH_4$ and $NO_x$, and hence the $NO_x$:$CH_4$ ratio for this dataset was calculated. Figure 9 shows the ECLIPSE $NO_x$:$CH_4$ emission ratio in the North Sea, in units of mass per unit mass.

Conversion of the $\Delta NO_x$:$\Delta CH_4$ ratio measured in this work (in units of mole fraction per unit mole fraction) yields a median $\Delta NO_x$:$\Delta CH_4$ of 0.63 g g$^{-1}$ (mean = 1.14 ($\pm$ 1.54) g g$^{-1}$). The measured values were roughly 30 times greater than the highest ECLIPSE ratios in the North Sea, although $NO_x$:$CH_4$ ratios in the ECLIPSE inventory globally reached values greater than 2.0 Gg Gg$^{-1}$ (see Appendix D). Our study finds that the ECLIPSE inventory may underestimate the $NO_x$:$CH_4$ ratio by more than an order of magnitude in the North Sea region.


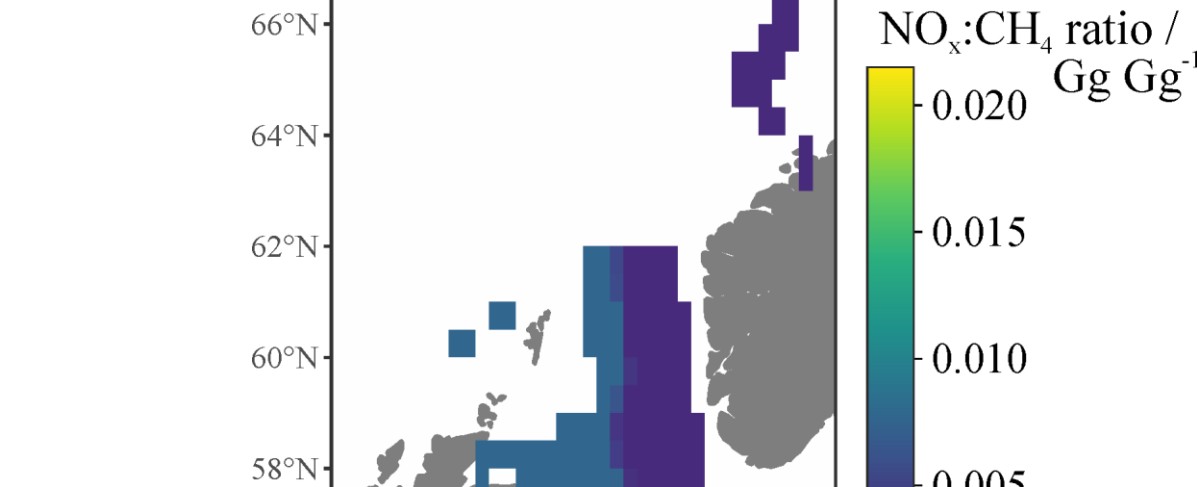

**Figure 9. ECLIPSE v5 $NO_x$:$CH_4$ ratios in the North Sea.**



## 4 Atmospheric implications

Flaring in the UK North Sea reportedly fell by 19% in 2021, but 740 million cubic metres ($7.4 \times 10^8$ m$^3$) of natural gas were still reported to have been flared (OGA, 2021). Here, we use the median gas composition of flared gas provided by BEIS for this region ($CH_4 = 0.845$, and $C_2H_6 = 0.085$), and the median DREs for $CH_4$ and $C_2H_6$ (calculated in Section 3.2) to estimate total emissions of $CO_2$, $CH_4$, and $C_2H_6$ from North Sea flaring. We estimate that flaring in the UK North Sea resulted in total emissions of 1.4 Tg yr$^{-1}$ $CO_2$, 6.3 Gg yr$^{-1}$ $CH_4$, and 1.7 Gg yr$^{-1}$ $C_2H_6$. Using the calculated $CH_4$ emission total here, and the

median $\Delta NO_x{:}\Delta CH_4$ ratio derived in Section 3.3, we estimate total emissions of 3.9 Gg yr$^{-1}$ $NO_x$ from flaring in the North Sea region. These values, estimated using reported flaring volumes and statistics measured as part of this work, can be compared against the total emissions estimated by inventories for the North Sea region. ECLIPSE reports 30 times greater emissions of $CH_4$ from the North Sea, with 177 Gg yr$^{-1}$ $CH_4$, but smaller emissions of $NO_x$ of 0.9 Gg yr$^{-1}$ $NO_x$. The lower $NO_x$ estimate is potentially the result of the lower $NO_x{:}CH_4$ ratio in the ECLIPSE model, which largely underestimated the $NO_x{:}CH_4$ ratio

relative to that measured in this work (Section 3.4). Alternatively, the Global Fuel Exploitation Inventory (GFEI) provides $CH_4$ emissions of 13.9 Gg yr$^{-1}$ $CH_4$, 3 times greater than our own estimate here for the North Sea region. The GFEI total can be broken down into 11.8 Gg yr$^{-1}$ $CH_4$ (85%) from flaring during oil exploitation, 1.5 Gg yr$^{-1}$ $CH_4$ (11%) from gas processing, and 0.5 Gg yr$^{-1}$ $CH_4$ (4%) from gas production. Neither inventory provided flaring emission products for $CO_2$ or $C_2H_6$, and GFEI did not include $NO_x$ flaring emissions. These results are summarised in Table 2.


**Table 2. Estimated total emissions of $CO_2$, $CH_4$, $C_2H_6$, and $NO_x$ from flared natural gas in the North Sea (in Gg), and globally (in Tg).**

| Data source | North Sea Flaring Emissions / Gg yr$^{-1}$ | | | | Global Flaring Emissions* / Tg yr$^{-1}$ | | | |
|---|---|---|---|---|---|---|---|---|
| | CO$_2$ | CH$_4$ | C$_2$H$_6$ | NO$_x$ | CO$_2$ | CH$_4$ | C$_2$H$_6$ | NO$_x$ |
| This work | 1400 | 6.3 | 1.7 | 3.9 | 245 | 5.6 | 1.1 | 3.6 |
| ECLIPSE[1] | | 177 | | 0.9 | | 109 | | 0.3 |
| GFEI[2] | | 13.9 | | | | 0.6 | | |
| IEA[3] | | | | | 265 | 8 | | |
| Plant et al. (2022) | | | | | | 7.6 | | |

\* Uses the DRE measured in this work for offshore flaring (25% of global total; IEA, 2018), and the DRE measured by Plant et al. (2022) for onshore flaring (75% of global total; IEA, 2018). [1] Stohl et al., 2015. [2] Scarpelli et al., 2020. [3] IEA, 2021.


Extrapolating the results of this work to the global scale relies on the crude assumption that global natural gas supplies are analogous to those found in the North Sea, and that operational practices are consistent across all fields and regions both onshore and offshore. Such an extrapolation could be useful even with these substantial assumptions, as measurements of





combustion efficiencies and $NO_x$ emission ratios from flared gas are exceptionally rare, especially offshore. Plant et al. (2022)

presented a global extrapolation of their own measured flaring efficiency, estimating global total emissions of 7.6 Tg $CH_4$ using an effective $DRE_{CH_4}$ (which includes measures of unlit flares) of 91.1%. The proportion of unlit flares was observed to be between 3% and 5% of all flares across different onshore basins in the United States (Lyon et al., 2021; Plant et al., 2022) and therefore may be significant for extrapolating total emissions. If we assume the $DRE_{CH_4}$ value measured by Plant et al. (2022) is appropriate for all onshore production, and that our own measured DRE values are appropriate for offshore

production, we can provide an alternative global extrapolation that accounts for any systematic differences between onshore and offshore flaring.

Approximately 25% of global oil and gas supplies are produced offshore (IEA, 2018). The IEA reported that 142 billion cubic metres ($142 \times 10^9$ m$^3$) of natural gas were flared worldwide in 2020 (IEA, 2021). If flaring is practiced to the same extent both onshore and offshore, then it follows that offshore flaring was responsible for approximately $36 \times 10^9$ m$^3$ of

the global total. By assuming that the median DREs calculated here and the median fuel gas composition values provided by BEIS for North Sea platforms are appropriate for offshore production globally, we estimate global offshore flaring emissions of 65 Tg yr$^{-1}$ $CO_2$, 0.3 Tg yr$^{-1}$ $CH_4$, and 0.08 Tg yr$^{-1}$ $C_2H_6$. Using the onshore measured effective DRE for $CH_4$ from Plant et al. (2022) for both $CH_4$ and $C_2H_6$, we estimate global onshore flaring emissions of 180 Tg yr$^{-1}$ $CO_2$, 5.3 Tg yr$^{-1}$ $CH_4$, and 1.0 Tg yr$^{-1}$ $C_2H_6$. Total global emissions, from both onshore and offshore flaring, were therefore 245 Tg yr$^{-1}$ $CO_2$, 5.6 Tg yr$^{-1}$ $CH_4$,

and 1.1 Tg yr$^{-1}$ $C_2H_6$. Our estimate of $CO_2$ emissions is consistent with the IEA estimate, but our estimate of $CH_4$ emission is lower. This is due to the higher combustion efficiency measured for the North Sea (median = 98.4%) and used for offshore estimates, compared to the lower estimate of 92% used by the IEA for both onshore and offshore flaring globally. Using the median $\Delta NO_x{:}\Delta CH_4$ ratio, flaring was estimated to be responsible for emissions of 3.6 Tg yr$^{-1}$ $NO_x$ globally. Comparing to the emission inventories, ECLIPSE provides much greater total annual emissions of $CH_4$, of 109 Tg yr$^{-1}$, but lower emissions of

$NO_x$, of 236 Gg yr$^{-1}$. GFEI provides total global $CH_4$ emissions of 630 Gg yr$^{-1}$, of which oil exploitation contributes 500 Gg yr$^{-1}$ (79%), gas processing 95 Gg yr$^{-1}$ (15%), and gas production 35 Gg yr$^{-1}$ (6%). Total global emissions of $CO_2$, $CH_4$, $C_2H_6$, and $NO_x$ are summarised in Table 2.

## 5 Conclusions

Fifty-eight plumes were identified as containing emissions likely to result from flaring of natural gas from offshore oil and gas

facilities in the North Sea. Combustion efficiency, the efficiency with which the flares convert carbon in the fuel gas into $CO_2$, was calculated for each of these plumes using two approaches; with and without accounting for $C_2H_6$ in the flare plume. The median combustion efficiency, of 98.4% (with $C_2H_6$) and 98.7% (without $C_2H_6$), was in agreement with the assumed value of 98% used by many emission inventories for flaring combustion efficiency. The linear relationship between combustion efficiencies calculated with and without $C_2H_6$ could be used to derive more accurate combustion efficiencies in the absence of

measurements of $C_2H_6$, assuming similar fuel gas composition. Destruction removal efficiencies (DREs) were also calculated





for $CH_4$ and $C_2H_6$ in each plume, making use of fuel gas compositions provided by BEIS. Median DRE values were 98.5% and 97.9% for $CH_4$ and $C_2H_6$ respectively.

$NO_x$ emission ratios were calculated using both $CO_2$ and $CH_4$ as reference gases, with median values of 0.003 and 0.26 ppm ppm$^{-1}$ for $CO_2$ and $CH_4$ as a reference respectively. All five of the greatest $\Delta NO_x:\Delta CH_4$ ratios (>1.1 ppm ppm$^{-1}$) and 465 $\Delta NO_x:\Delta CO_2$ ratios (>0.011 ppm ppm$^{-1}$) were measured in the vicinity of Floating Production Storage and Offloading vessels, which may indicate a difference in their flaring operation compared with fixed platforms. $C_2H_6$ emission ratios were calculated using $CH_4$ as a reference gas. The median value for $\Delta C_2H_6:\Delta CH_4$, of 0.11, was in excellent agreement with $C_2H_6$ emission ratios calculated for similar datasets. Wind speed appeared to have only a small impact on both the combustion efficiency of the flares, and the relative amount of $NO_x$ produced, although more data on flares operating in wind speeds of greater than 15 470 m s$^{-1}$ are needed.

Total North Sea and total global emissions due to flaring were estimated using reported gas flaring volumes and the statistics calculated in this work. For the North Sea, emissions were estimated as 1.4 Tg yr$^{-1}$ $CO_2$, 6.3 Gg yr$^{-1}$ $CH_4$, 1.7 Gg yr$^{-1}$ $C_2H_6$, and 3.9 Gg yr$^{-1}$ $NO_x$, whilst globally emissions were extrapolated to 245 Tg yr$^{-1}$ $CO_2$, 5.6 Tg yr$^{-1}$ $CH_4$, 1.1 Tg yr$^{-1}$ $C_2H_6$, and 3.6 Tg yr$^{-1}$ $NO_x$. Although many emission inventories do include emissions from flaring, most do not provide separate 475 values for this source, and instead aggregate emissions due to flaring with other oil and gas sector emissions. This makes comparison challenging. However, we find that the ECLIPSE inventory overestimates $CH_4$ emissions from flaring by a factor of 30 in the North Sea, but underestimates $NO_x$ emissions by a factor of 4. The GFEI product overestimates $CH_4$ emissions from flaring by a factor of 2 in the North Sea.

The skewed distribution of combustion efficiencies found in this, and other, studies indicates that many flares operate 480 below the assumed standard efficiency for combustion. Inefficient combustion, together with the prevalence of unlit flares which directly vent $CH_4$ to atmosphere, contribute to large $CH_4$ emissions. Hence, improving natural gas disposal and flaring practices represents a viable strategy for mitigating carbon emissions from the oil and gas sector.



## Appendix A: Impact of data availability on plume exclusions


**Table A1. Data availability (percentage of total 1 Hz data) for $CO_2$, $CH_4$, $C_2H_6$, and $NO_x$ during FAAM AEOG and MOYA flights. Data availability below 50% are highlighted in red. It should be noted that 100% data availability would not be expected for various reasons. Firstly, data files might contain data outside of when the instruments were operational (e.g. before take-off, or after landing) which were removed for analysis; and secondly, due to the presence of instrument calibrations, for which data were flagged and**

**removed.**

| Flight No. | $CO_2$ data / % | $CH_4$ data / % | $C_2H_6$ data / % | $NO_x$ data / % |
|---|---|---|---|---|
| C099 | 87 | 87 | 53 | 56 |
| C100 | 83 | 83 | 39 | 18 |
| C102 | 88 | 88 | 53 | 53 |
| C118 | 83 | 83 | 31 | 3.0 |
| C119 | 83 | 83 | 50 | 40 |
| C120 | 86 | 86 | 17 | 6.0 |
| C121 | 84 | 84 | 29 | 50 |
| C147 | 92 | 92 | 13 | 20 |
| C148 | 94 | 94 | 50 | 2.7 |
| C149 | 93 | 93 | 17 | 39 |
| C150 | 95 | 95 | 32 | 3.5 |
| C151 | 95 | 95 | 23 | 22 |
| C191 | 89 | 89 | 72 | 58 |
| C193 | 90 | 90 | 74 | 25 |





**Table A2. Reasons for plume exclusion. See Section 2.3 for detailed criteria descriptions. Note that plumes could be excluded based on failing multiple criteria.**

| Component | Background values < 10 | Within-plume values < 3 | Low maximum enhancement < 2σ above background |
|---|---|---|---|
| $CH_4$ | 0 | 0 | 0 |
| $CO_2$ | 0 | 0 | 1 |
| $NO_x$ | 44 | 11 | 2 |
| $C_2H_6$ | 4 | 9 | 7 |



**Appendix B: Comparing results for plumes with the same source origin**

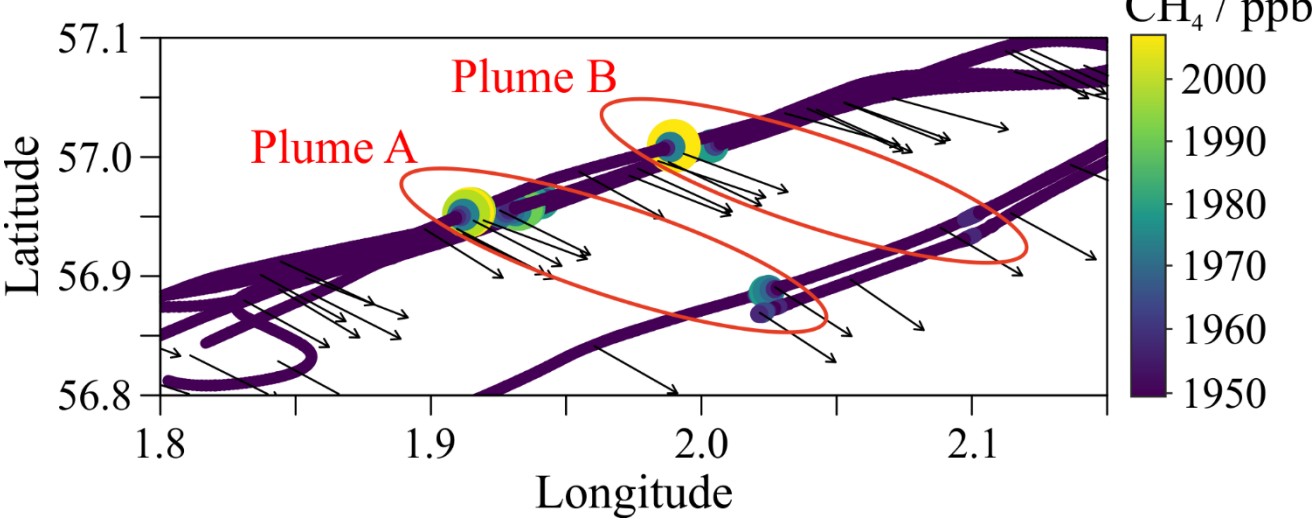

**Figure B1. CH₄ mole fraction (see colour scale) measurements in the North Sea on 4 March 2019. Black arrows show the 60 second mean wind direction. Two distinct emission plumes (containing enhancements in CH$_4$ as well as CO$_2$, NO$_x$, and C$_2$H$_6$) are shown, labelled Plume A and Plume B. Note that some of these peaks were removed from analysis due to a lack of measured data (primarily NO$_x$) either within plume or within the background (see Appendix A).**





**Table B1. Combustion efficiencies (with and without $C_2H_6$) and emission ratios for peaks within two plumes sampled on 4 March 2019 (see Fig. B1).**

| Plume 1 | | | | | | | | |
|---|---|---|---|---|---|---|---|---|
| Time | Latitude | Longitude | Wind speed / m s$^{-1}$ | Combustion efficiency (without $C_2H_6$) | Combustion efficiency (with $C_2H_6$) | $NO_x$:$CO_2$ | $NO_x$:$CH_4$ | $C_2H_6$:$CH_4$ |
| 14:07 | 56.96 | 1.94 | 15.3 | 95.4 | 94.5 | 0.0018 | 0.038 | 0.111 |
| 14:14 | 56.96 | 1.94 | 19.6 | 95.8 | 95.0 | 0.0021 | 0.047 | 0.107 |
| 14:19 | 56.95 | 1.93 | 15.2 | 97.4 | 96.8 | 0.0024 | 0.087 | 0.113 |
| 14:26 | 56.96 | 1.93 | 16.1 | 96.9 | 96.2 | 0.0020 | 0.062 | 0.110 |
| 14:33 | 56.96 | 1.92 | 15.6 | 97.6 | 97.1 | 0.0022 | 0.090 | 0.106 |
| Average | | | 16.4 ± 1.8 | 96.6 ± 0.9 | 95.9 ± 1.1 | 0.0021 ± 0.0002 | 0.065 ± 0.023 | 0.109 ± 0.003 |
| Plume 2 | | | | | | | | |
| 14:08 | 57.01 | 2.00 | 13.9 | 97.8 | 97.3 | 0.0025 | 0.11 | 0.095 |
| 14:13 | 57.01 | 2.00 | 17.1 | 99.2 | 99.1 | 0.0036 | 0.46 | 0.100 |
| 14:21 | 57.01 | 2.00 | 13.4 | 98.6 | 98.3 | 0.0031 | 0.22 | 0.124 |
| 14:34 | 57.01 | 2.00 | 15.1 | 98.0 | 97.5 | 0.0026 | 0.13 | 0.123 |
| 14:52 | 56.95 | 2.10 | 16.3 | 98.7 | 98.4 | 0.0023 | 0.18 | 0.134 |
| Average | | | 15.2 ± 1.5 | 98.5 ± 0.6 | 98.1 ± 0.7 | 0.0028 ± 0.0005 | 0.22 ± 0.14 | 0.115 ± 0.017 |





**Appendix C: Additional data presentation**

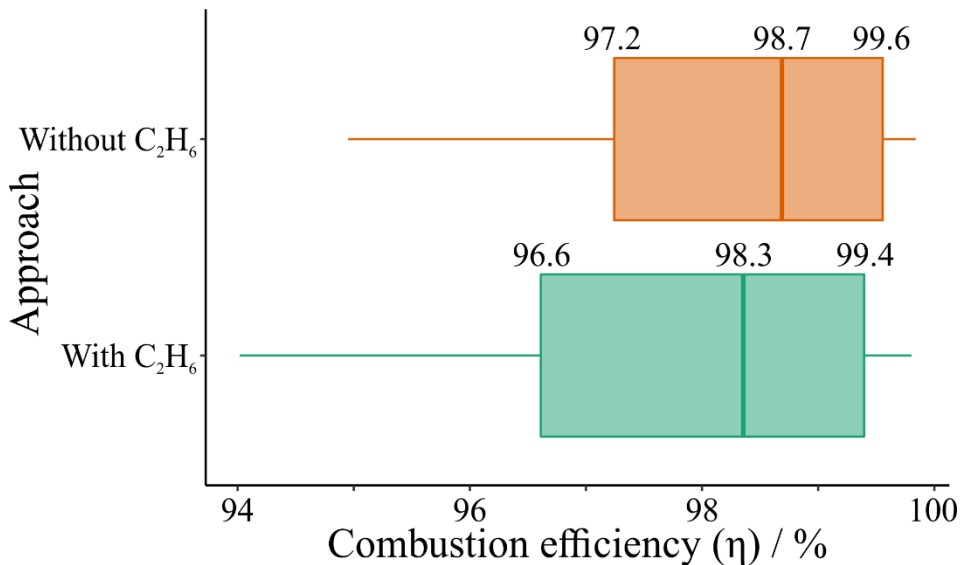

**Figure C1. Box and whisker plots of combustion efficiencies calculated without C₂H₆ (Eq. 2; orange, top row) and with C₂H₆ (Eq. 3; green, bottom row).**

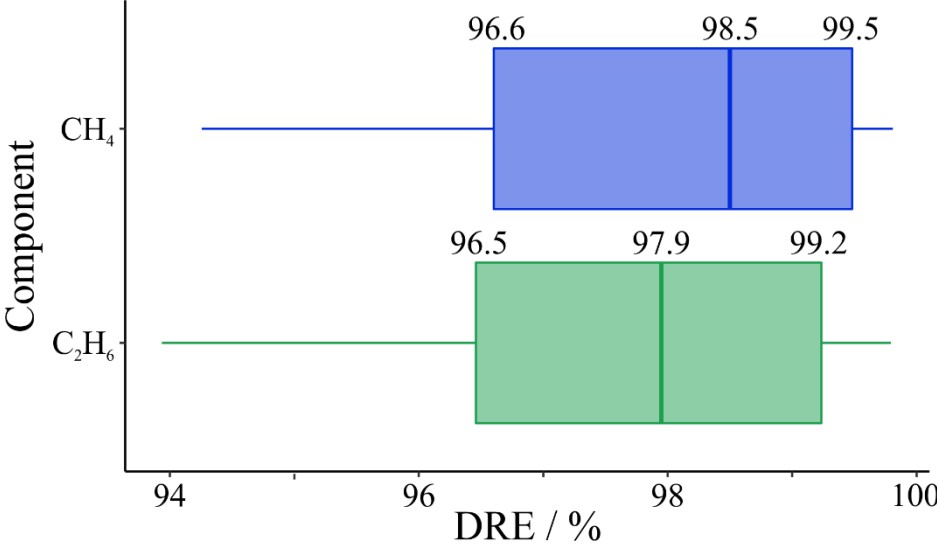

**Figure C2. Box and whisker plots of destruction removal efficiencies (DREs) for CH₄ (blue; top row) and C₂H₆ (green; bottom row).**



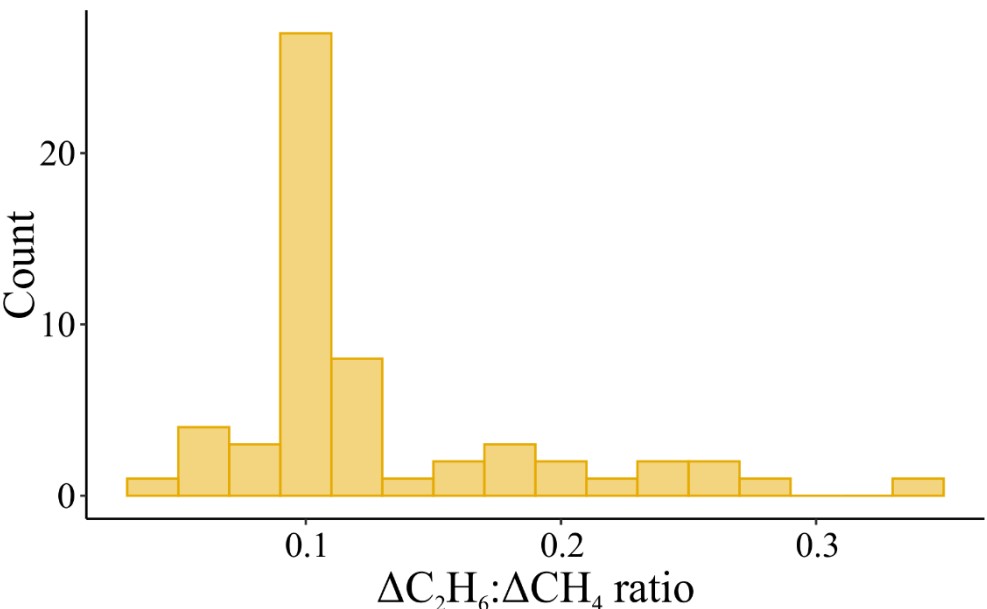

**Figure C3. Histogram distribution of $\Delta C_2H_6$:$\Delta CH_4$ ratios.**




**Appendix D: Flaring emissions inventory maps (global and North Sea)**

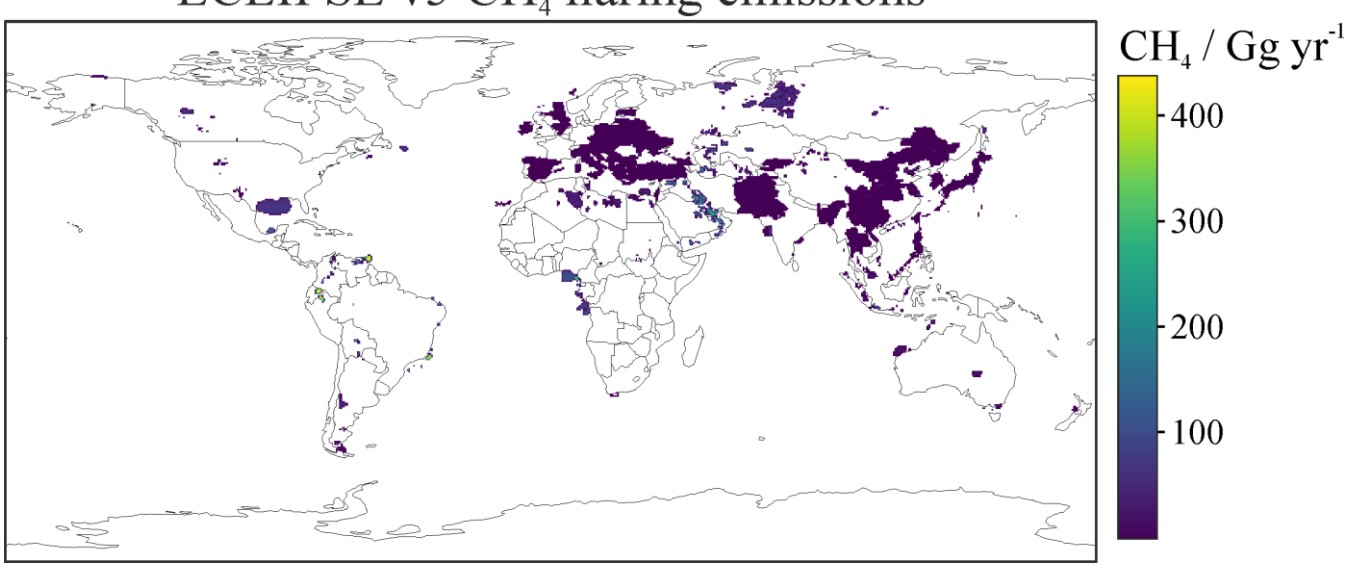


**Figure D1. ECLIPSE v5 global CH$_4$ flaring emissions at 0.5° × 0.5° for 2020 (Stohl et al., 2015).**

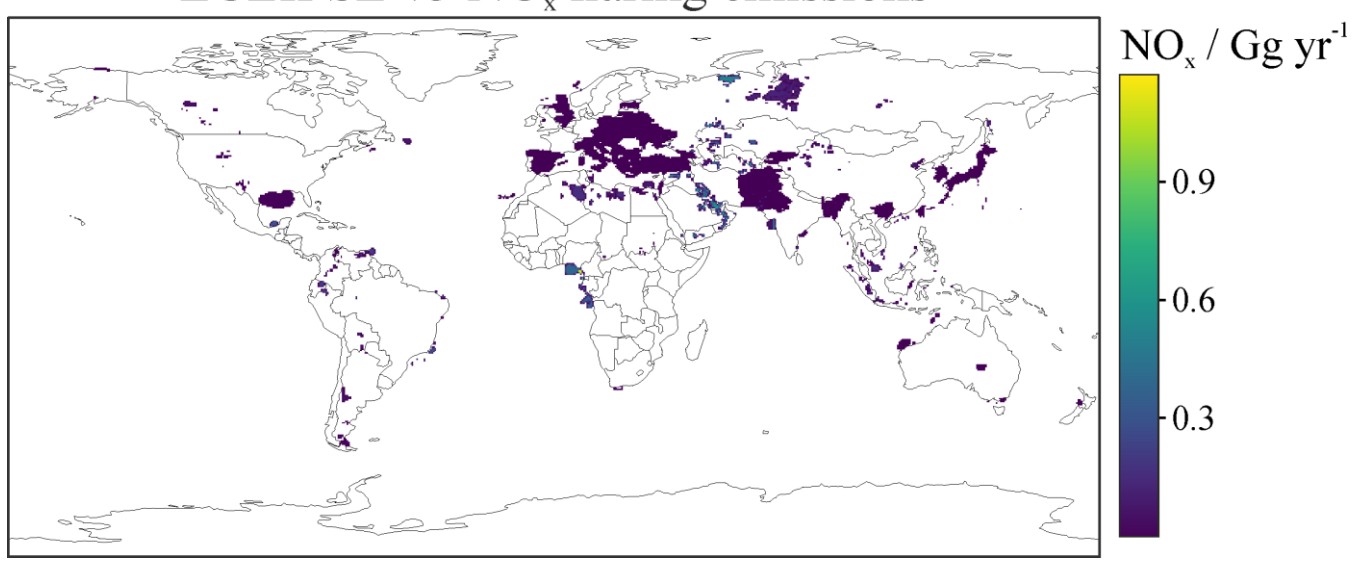

**Figure D2. ECLIPSE v5 global NO$_x$ flaring emissions at 0.5° × 0.5° for 2020 (Stohl et al., 2015).**




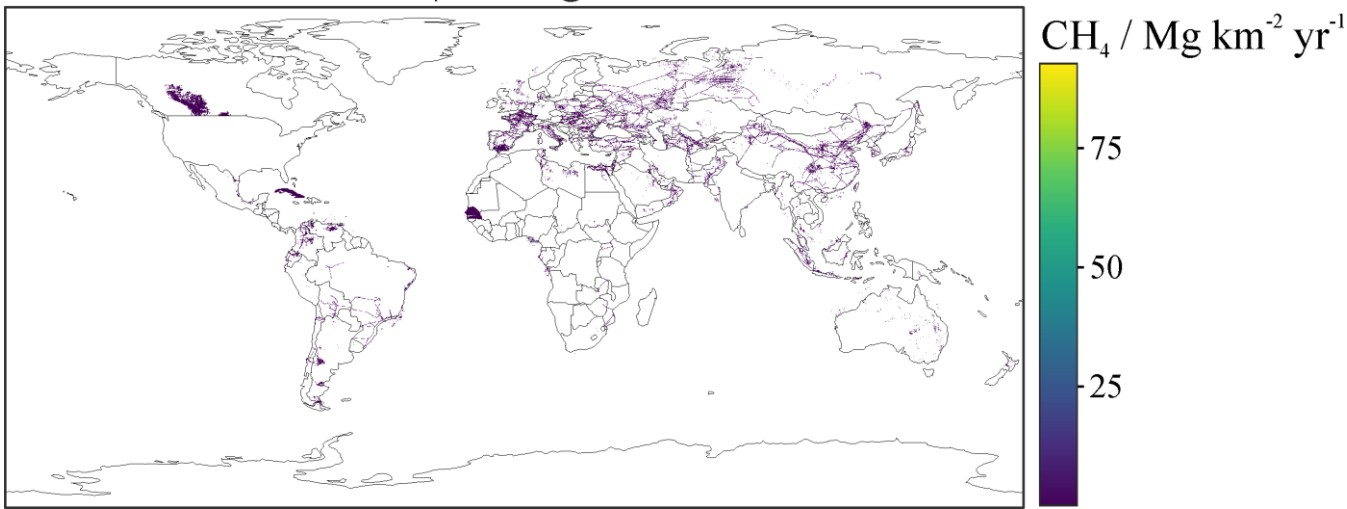

**Figure D3. GFEI global CH₄ flaring emissions at 0.1° × 0.1° for 2019 (Scarpelli et al., 2020).**

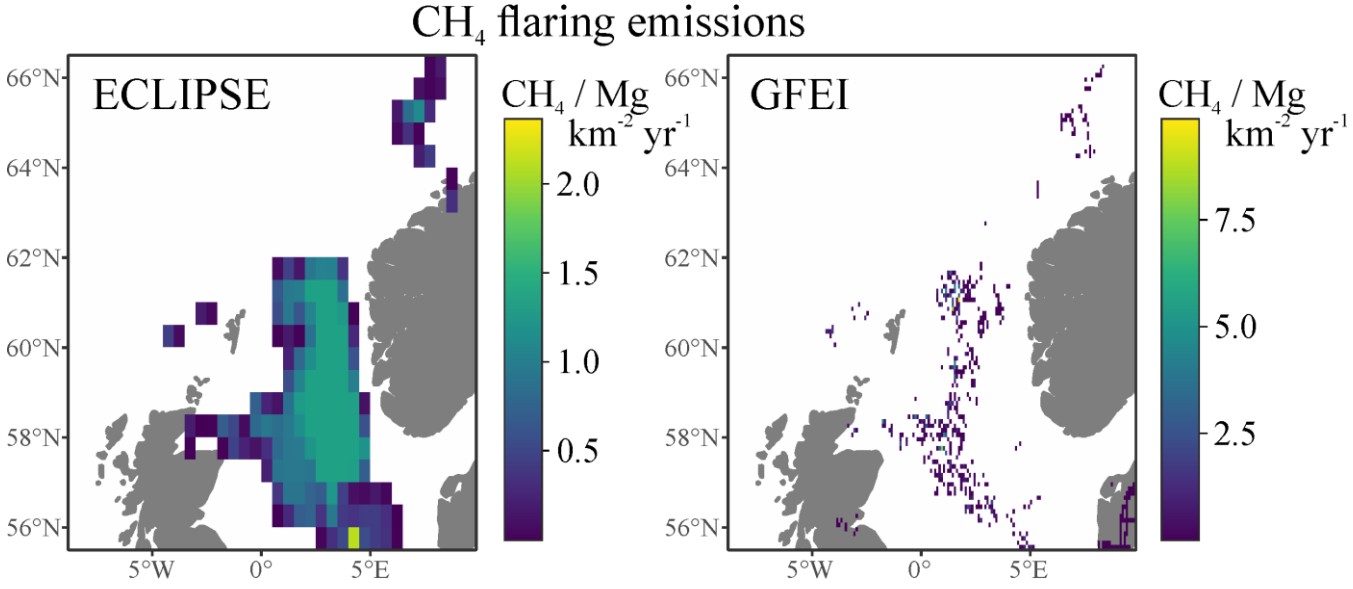

**Figure D4. Left) ECLIPSE v5 CH₄ flaring emissions over the North Sea, at 0.5° × 0.5° for 2020 (Stohl et al., 2015). Right) GFEI CH₄ flaring emissions over the North Sea, at 0.1° × 0.1° for 2019 (Scarpelli et al., 2020).**





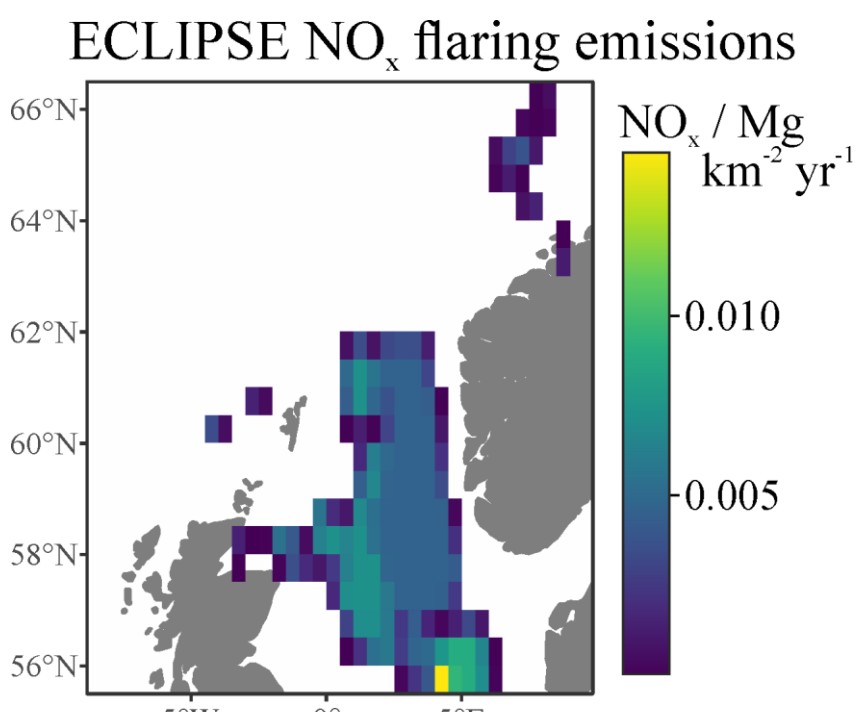

**Figure D5. ECLIPSE v5 NO_x flaring emissions over the North Sea, at $0.5° \times 0.5°$ for 2020 (Stohl et al., 2015).**


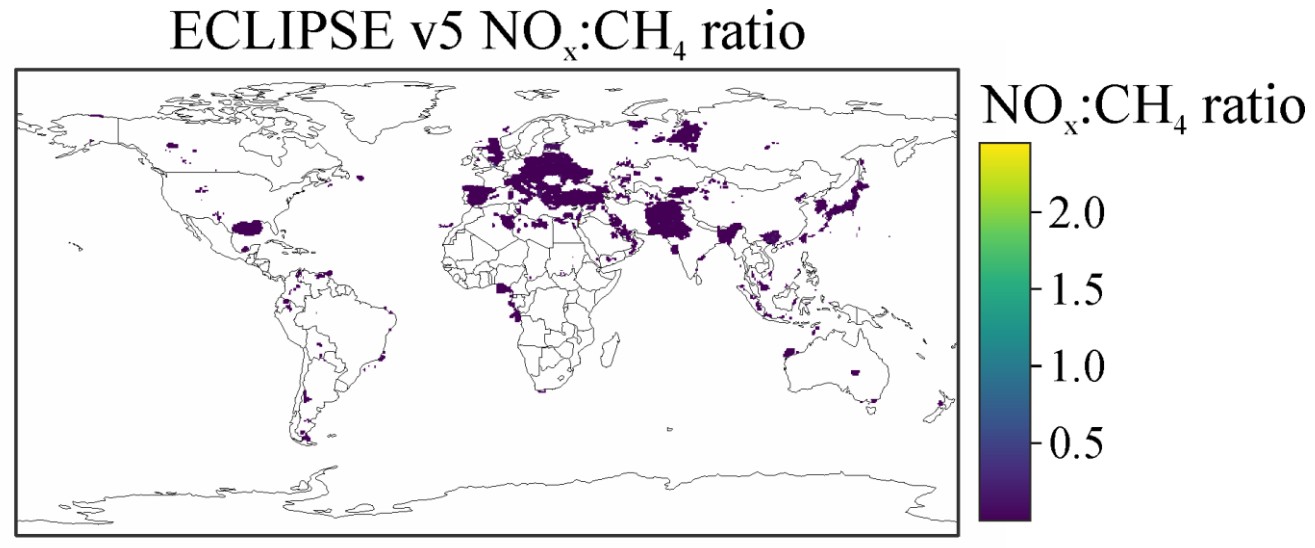

**Figure D6. ECLIPSE v5 NO_x:CH_4 ratio, at $0.5° \times 0.5°$ for 2020 (Stohl et al., 2015).**



**Appendix E: VIIRS data**

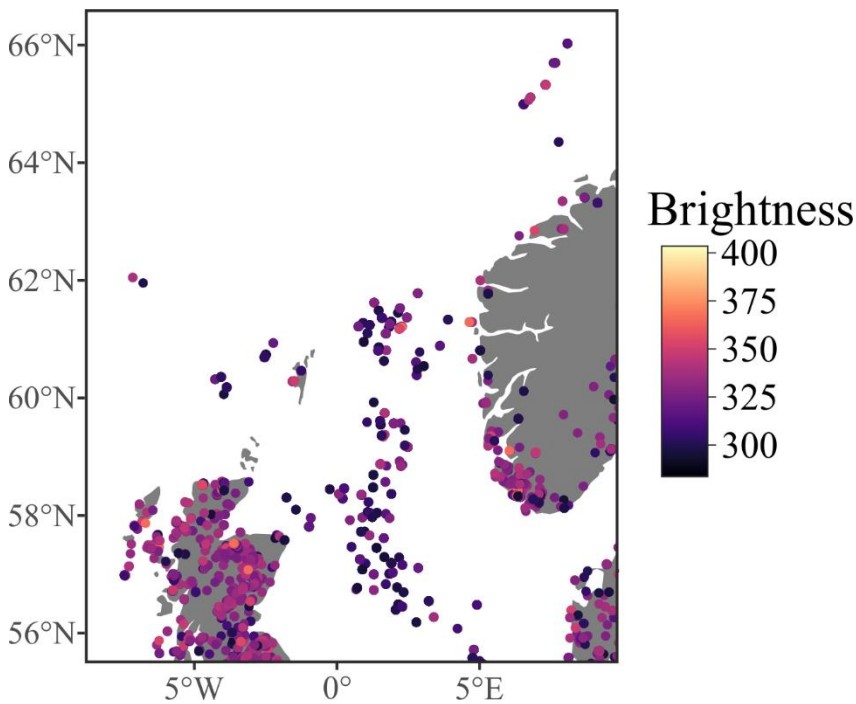


**Figure E1. Locations of active fire and thermal anomalies measured by the Moderate Resolution Imaging Spectroradiometer (MODIS) and Visible Infrared Imaging Radiometer Suite (VIIRS) Suomi-NPP between 01 September 2018 and 31 August 2019. Note that these data show the locations of observed thermal anomalies, which may or may not be indicative of natural gas flaring, although those over the North and Norwegian Seas are expected to be so. MODIS Collection 61 NRT Hotspot / Active Fire Detections**
**MCD14DL distributed from NASA FIRMS. Available online https://earthdata.nasa.gov/firms; https://doi.org/10.5067/FIRMS/MODIS/MCD14DL.NRT.0061NRT. VIIRS 375 m Active Fire product VNP14IMGT distributed from NASA FIRMS. Available online https://earthdata.nasa.gov/firms; https://doi.org/10.5067/FIRMS/VIIRS/VNP14IMGT_NRT.002.**




**Data availability**

Data from the AEOG and MOYA FAAM aircraft campaigns are available from the Centre for Environmental Data Analysis (CEDA) archive (https://www.ceda.ac.uk) at https://catalogue.ceda.ac.uk/uuid/c94601501623483aa0a12e29ce99c0e0 (Crosier, 20220) and https://catalogue.ceda.ac.uk/uuid/dd2b03d085c5494a8cbfc6b4b99ca702 (Nisbet, 2022) respectively.
Please note that access to CEDA data sets and resources requires a free CEDA login account. This is in-line with funder policy and ensures appropriate use and citation of public data. GFEI emission grids are available for download from the Harvard Dataverse at https://doi.org/10.7910/DVN/HH4EUM (Scarpelli and Jacob, 2021). ECLIPSE global emission grids based on the GAINS model are publicly available from https://previous.iiasa.ac.at/web/home/research/researchPrograms/air/Global_emissions.html (IIASA, 2015).

**Author contributions**

**JTS** – Formal analysis, Methodology, Visualization, Writing – original draft preparation; **AF** – Formal analysis, Methodology, Writing – original draft preparation; **SW** – Formal analysis, Investigation, Visualization, Writing – original draft preparation; **PB** – Data curation, Investigation; **FS** – Data curation, Investigation; **JL** – Conceptualization, Investigation, Project administration, Funding acquisition; **RP** – Conceptualization, Investigation, Funding acquisition; **RB** –
Investigation, Funding acquisition; **IC** - Investigation; **SM** – Investigation, Funding acquisition; **SJBB** – Data curation, Investigation; **SY** – Data curation, Investigation; **SS** – Writing – original draft preparation; **GA** – Conceptualization, Investigation, Methodology, Project administration, Writing – original draft preparation; Funding acquisition.

**Competing interests**

The contact author has declared the neither they nor their co-authors have any competing interests.

**Acknowledgements**

This work was supported by the Climate and Clean Air Coalition (CCAC) Oil and Gas Methane Science Studies (MSS) hosted by the United Nations Environment Programme. Funding was provided by the Environmental Defense Fund, the Oil and Gas Climate Initiative, the European Commission, and CCAC (Grant No.: DTIE19-020). The aircraft data used in this publication were collected as part of two projects: the Demonstration Of A Comprehensive Approach To Monitoring
Emissions From Oil and Gas Installations (AEOG) project (Grant No.: NE/R01451X/1), and the Methane Observations and Yearly Assessment (MOYA) project (Grant No.: NE/N015835/1), both funded by the Natural Environment Research Council (NERC). We would like to thank Airtask Ltd. (who flew the aircraft), and all those involved in the operation and maintenance of the BAe-146-301 Atmospheric Research Aircraft, including FAAM, Avalon Aero, UK Research and Innovation (UKRI), and the University of Leeds. We also acknowledge the Offshore Petroleum Regulator for Environment
and Decommissioning (OPRED) and Ricardo Energy & Environment for their involvement as project partners on the AEOG project. Any opinions, findings, conclusions, or recommendations expressed in this material are those of the author(s) and do not necessarily reflect the views of their respective institutions.

**Financial support**

This research was supported by the Natural Environment Research Council (Grant No.: NE/R01451X/1 and NE/N015835/1) and the Climate and Clean Air Coalition (Grant No.: DTIE19-020).



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
