# Peer review of "Flaring efficiencies and $NO_x$ emission ratios measured for offshore oil and gas facilities in the North Sea"

_Atmospheric Chemistry and Physics, 2022_

## Author Comment (AC1)

**Authors' response to reviewer comments on ACP-2022-679**

The authors would like to thank the reviewers for their supportive comments. Reviewer comments are collated below in *italics* with corresponding author responses in blue text. Additions or amendments to the original manuscript are underlined.

Please note that we have made additional edits to add further context on flaring emissions and regulatory practices. Substantial edits (i.e., not just grammatical), that were not a direct result of the reviewer comments below, are collated at the end of this document for transparency.

Reviewer 1

*The authors present analysis of emissions efficiencies and ratios for four different gases from North Sea oil and gas flaring. They have used data from two flight campaigns and measured 58 flaring plumes using the FAAM aircraft between 2018 and 2019. By using combustion efficiency and destruction removal efficiency calculations the authors show that previous assumptions of emissions from this sector broadly agree with field measurements. The calculated combustion efficiencies are between 94 – 100% and the authors demonstrate that including ethane in combustion efficiency calculations has a very small impact on the final result. They also show no statistical correlation between wind speed and combustion efficiency but higher combustion efficiencies were measured in the Norwegian sector of the region. Comparisons with two emission datasets show their calculated emission ratios to be 30 times greater than that of the ECLIPSE inventory resulting in 30 times less methane emissions when compared to the emission dataset. They conclude the paper by extrapolating their results to a global scale, based on coarse assumptions.*

*I think this is generally a good paper and is worth publication. The results presented are not ground breaking or particularly controversial but are non the less important to back up assumptions of emissions from flaring.*

We would like to thank the reviewer for their time reading and reviewing our publication. We agree that our results are important measurement comparisons with the assumptions made in inventories and emissions accounting but would like to reiterate that the results *are* novel in the context of offshore flaring emissions. We believe these to be the first measurements of offshore flaring emissions, which may differ from onshore flares in both design and function.

*Line 37 – This would be more readable as 142 billion rather than 149 x 10^9*

We agree and this has been amended as suggested.

*Line 73 – I think 'shipborne-based' should either be shipborne, or ship-based*

The reviewer is correct. This has been amended to read "shipborne".

*Line 74 – I do not understand why the sentence about isotopic ratios is important here as there is nothing about it in the rest of the paper. What is the context of this statement?*

The sentence referred to reads: "The carbon isotopic signature of methane emitted from oil and gas facilities is useful for source identification and has been measured to be -53‰ in the North Sea."

This paragraph in the introduction provides context on recent measurements of oil and gas emissions in the North Sea, collating results from various studies estimating emissions of both

methane and VOCs. The sentence in question mentions that isotopic studies have also been conducted. As the reviewer correctly states, isotopic signatures are not part of this work. However, we believe that this sentence can be retained as part of a brief overview of recent measurements relating to oil and gas emissions in the North Sea.

*Figure 1 – Lat/lon grid lines would be useful on this map to be able to get a sense of scale when comparing to the model resolution*

We agree with the reviewer and have amended Fig. 1 as below to include latitude and longitude grid lines:

[Figure]

*Section 2.3 – It is unclear how you decide what/where a plume is in the data. It seems as if you decide a plume if its elevated from the background, and decide the background if its not a plume which seems a bit circular. 50 neighbouring measurements either side of a plume seems ok but clarification on if there is a measurement limit for when the plume starts and stops is needed here.*

In this work, we use the same plumes as those identified by Foulds et al. (2022) – please refer to this for more information. Briefly, peaks/enhancements indicative of plumes were initially (and roughly) identified by manually examining time series of the $CH_4$ and $CO_2$ data. A background (and associated standard deviation) was then identified for each flight survey (manifested as a mode in a histogram with a value approximately 2 ppm – roughly equivalent to the northern hemisphere $CH_4$ background). Plume start-times were defined where the $CH_4$ concentration exceeded 2 standard deviations above the background value, and plume end-times were defined where the $CH_4$ concentration dropped to within 2 standard deviations of the background value. All plumes identified in this way were checked against those identified through manual examination. This was done to ensure that actual plumes were identified, and not just singular (or extreme) data points in the time-series.

After identifying plumes, a plume-specific background was calculated using the mean of the 50 nearest neighbouring datapoints to either side of the plume (where this data was within 2 standard deviations of the flight background). This allowed for local variations in the $CH_4$ background

to be accounted for. In practice, this method works very well in "clean" environments, such as those over the ocean, where background variability is low.

However, we recognise that there may be some data points which do not contribute to the enhancement (between the mean plume-background and the flight-background + 2 standard deviations). To mitigate this, background uncertainty is included in our error budget via the inclusion of an uncertainty term in calculating enhancements.

We have added the following brief description to the paper:

"Emissions from oil and gas facilities were identified in flight time-series data using the method described in Foulds et al. (2022). Briefly, plumes were both manually and statistically identified. Manual identification relied on visual inspection of the time-series data for enhancements. Statistical identification involved the determination of a background (and associated standard deviation) for each flight survey, manifested as a mode in the data of approximately 2 ppm $CH_4$ (equivalent to the northern hemisphere $CH_4$ background). Emission plumes were defined as enhancements that exceeded two standard deviations about the flight-specific background value. Manually and statistically identified plumes were compared to confirm likely emissions and not just singular, extreme data points in the time-series."

*Lines 210 - 216 – What is the variation of fuel composition data you do have? Is the median value representative or is there a large spread of values and do you have an indication on whether this value would impact the results in major way or not?*

It should be noted that there were only 15 peaks for which fuel composition data was unavailable, so this issue only affected roughly 25% of results.

There was a large range in provided fuel composition data. The composition of methane ranged from 31% to 97%, with a mean of 74% ± 22%. For ethane, the composition data ranged from 2.2% to 18%, with a mean of 8.7% ± 4.9%. A Monte Carlo sensitivity simulation (n = 10,000) showed that this range of compositional values gave less than 1% uncertainty (1σ) to the calculated DREs for both methane and ethane. The manuscript has been updated to read: "A Monte Carlo simulation (n = 10,000) showed that calculated DREs were not sensitive to the choice of composition value, with a less than 1% uncertainty (1σ) in mean DREs across the distribution of provided composition values."

*Section 3.4 – It need to be clearer as to why the large disparity between ECLIPSE inventory and the observations is not a resolution problem. One the face of it would seem that a few pinpoint measurements in a 0.5 x 0.5 degree inventory is never going to match but I think (after a bit of head scratching) that it shouldn't matter. But it would be useful to have more explanation so the reader doesn't have to work this out themselves. Are your measurements going to be representative of the large grid square?*

In Section 3.4 we compared the $NO_x$:$CH_4$ ratios in the ECLIPSE inventory against our own measured ratios (in g g$^{-1}$). The ECLIPSE inventory ratios ranged between 0 and 0.02 (mass per unit mass), whilst our measured ratios ranged between 0.1 and 10 (mass per unit mass), indicating a substantial disparity between the two datasets. Our measurements of $NO_x$:$CH_4$ ratio are representative of an inventory grid cell only if we sampled a representative population of flaring emissions within that grid cell. There are several factors which could contribute to the disparity here: firstly, the snapshot nature of our measurements mean that our ratios are representative of a limited timeframe of flaring. Emissions inventories are typically aggregated annually. If flaring emissions were to vary with time-of-year, (due to e.g., different flaring requirements or operations),

this may lead to discrepancies between the annual inventory and our snapshot measurements. The following has been added to Section 3.4:

"There are a few possible reasons for this disparity in NOx:CH4 ratios between datasets. Firstly, inventories are typically representative of annual emissions, whereas our ratios are 'snapshots' calculated for emissions at the time of sampling. If flaring emissions can be expected to vary throughout the year, either as a result of changes to operation or to local meteorology, this may lead to differences. Secondly, our measurements are only comparable to inventory grid cells if a representative population of flaring emissions were sampled. Thirdly, the ECLIPSE inventory for 2020 was calculated by projecting activity data for 2010 forwards in time using legislative and Representative Concentration Pathways (Klimont et al., 2017), and these may not be valid for current emission scenarios."

*Appendix figure D1, D2 & D3 – I don't think these maps are useful. The regional ones are but the global ones don't provide any useful information for this particular study.*

These maps were added in for additional global context on the range of values within the inventory. As the reviewer says, they do not provide meaningful information to the context of this work (which is focussed on the North Sea), and hence we have removed them as suggested.

*Appendix E – I may have missed this but where is this VIIRS data referred to in the text? Is this data used at all?*

The reviewer is correct: this plot is superfluous to this study and is not referenced in the text. The plot has been removed from the appendices.

Reviewer 2

*The authors present flaring efficiencies and emission ratios for 58 plumes measured during aircraft campaigns to investigate emissions from hydrocarbon production in the North Sea. They present combustion efficiencies, methane and ethane destruction removal efficiencies, and NOx emission factors. They find their estimates are roughly consistent with results from previous flaring studies, of which there are few. There has been little previous assessment of offshore flaring based on in-situ measurements. As a result, I find the work to be an important contribution to further the understanding of the full climate and air quality impacts of flaring during hydrocarbon extraction. The manuscript is well written and logically presented, however, there are few areas that I believe require further context and/or clarification. I detail these areas in my comments below.*

We would like to thank the reviewer for their time and effort in providing these comments, and for their supportive review.

*Lines 18-20: The authors provide combustion efficiency both with and without ethane. Is one thought to more accurate than the other?*

Combustion efficiency is defined as a measure of the efficiency with which a flare converts all carbon (i.e., any compound containing carbon) into carbon dioxide. Natural gas is mostly methane but generally contains some proportion of larger alkanes (like ethane), as well as other hydrocarbons. However, measuring many of these compounds at a high sampling rate (~1 Hz) is difficult (especially on aircraft). Hence, many previous (real-world) measurements have typically assumed all the carbon in the flare gas is present as $CH_4$. By necessity, this underestimates the amount of carbon in the flare gas (other hydrocarbons have more than one carbon atom), leading to

inaccurate estimates of combustion efficiency. Including ethane (usually the second highest fraction of flare gas) measurements in the total carbon for the flare gas should improve the accuracy of the combustion efficiency relative to just using methane. However, unless all hydrocarbons can be fully accounted for (i.e., measured), we will always only estimate combustion efficiency.

*Lines 119-120: 'However, as we used enhanced C2H6 mole fractions (background subtracted) in this work, the systematic altitude-dependent biases were effectively removed,' How does the use of enhanced C2H6 remove the altitude artifacts? Or, are the enhanced C2H6 mole fractions measured over a constant altitude, making the altitude-dependent biases irrelevant?*

Measurements of ethane were found to have a small dependency on altitude (due to optical effects in the instrument – see Pitt et al. 2016). Calibrations were not performed at every aircraft altitude due to time constraints. Hence, fully calibrated and quality assured data was only obtained at the altitudes at which the calibrations were performed.

Data at altitudes which were not calibrated for this altitude effect were therefore of a lower (reduced) quality. It should be noted that the calibration scheme should still calibrate the raw data for other instrument effects (e.g., temporal drift in instrument response), just not effects arising as a direct result of changes in altitude. It is expected that this altitude effect remains constant at a constant altitude. Therefore, measurements of ethane background ($C_{bg}$) and in-plume ethane ($C_{plume}$) at the same altitude will be subject to the same small effect, or bias ($E_{alt}$). Calculating ethane enhancement ($C_{enh}$) by subtracting one value from the other removes the impact of this (small) effect so long as it is constant at altitude.

$$\left(C_{plume} \pm E_{alt}\right) - \left(C_{bg} \pm E_{alt}\right) = C_{enh}$$

*Line 128: What constitutes a 'small temporal' discrepancy? <1s? 10s? If it is large, is there a chence the plumes might be misaligned for other reasons?*

The small temporal discrepancies were always less than 10 s (and mostly less than 5 s), and typically grew larger later in the day. This indicates a drift in the computer timing systems for the instruments, as they were only synced at the beginning of each day (this was incorrectly referred to as the start of each flight in the original manuscript). It should be noted that there will also likely be temporal discrepancies from instrument response times, and from the lengths of sampling lines between inlet and analyser for each instrument.

The manuscript has been corrected to read: "All instrumentation on board the FAAM aircraft were synchronised with respect to time  at the beginning of each day. However, instrument-specific temporal drift led to small temporal discrepancies (<10 s) …".

*Lines 149-157: The existence of correlated enhancements are used to select the flare plumes. Are expected signals such that you would be sure to see them given your instrument detection limits? Put another way, would it be possible that there are small signals you cannot see, and would this potentially bias your results to only larger flares?*

Yes, there is the possibility that the limits of detection (or rather, the instrument precision) on the instruments used here could mean that extremely small sources were missed. However, for them to be missed, they would have to be indistinguishable from the measured background (within 2 standard deviations of the estimated flight background – see response to comments above). Any emission sources within this range would be negligible when compared with the much stronger emission sources typically observed.

Fundamentally, any extremely small sources would be impossible to discern from the natural variability in the background airmass even with an instrument with perfect measurement precision. It would be impossible to ever measure them in practice using the sampling strategy in use here. We have included the following sentence to capture this:

"There is the potential that very small sources (with peak concentration enhancements less than two standard deviations in excess of the flight-specific background value) are not captured in this analysis. Such sources are indistinguishable from natural background variability and therefore cannot be accounted for."

*Line 158 (and throughout): Why do you use the median and not the mean here and throughout your analysis? It does not seem wrong but is there a reason why you do not use the mean?*

The flaring efficiency values were not normally distributed, in the same way that methane emission estimates from oil and gas are not normally distributed. Both distributions show a log-normal distribution, with a long-tailed skew of extreme values. In this case, using the mean-average would bias the results towards that extreme. Using a median average may not be a perfect solution, but hopefully better accounts for the higher proportion of values with less extreme values.

*Line 164: What is 'enough' data?*

In the case of a peak, 'enough' data was described as three points in the text – a central maximum value, and two values either side to demonstrate the width of the peak. In practice, this would be a poorly measured plume, but the requirement of three datapoints was set as an absolute lower limit. Only three plumes were excluded for solely having fewer than three datapoints within the plume – all other plumes that failed this criterion failed on multiple criteria.

*Line 192: The calculation of combustion efficiency here assumes the fuel is 100% CH4 and no CO2 is present in the fuel gas. Later you say the gas is on average ~85% CH4, so how does this assumption of 100% CH4 affect your results. You say there is a 'slight overestimation' but what is slight? Some sort of test case would help provide context here.*

'Slight' unfortunately depends entirely on the composition of the flare gas, and the destruction removal efficiency of each other hydrocarbon. The closer the true composition of methane is to 100%, the smaller the overestimation in combustion efficiency.

The below table gives an example scenario with a (fabricated) gas composition of 85% methane, 10% ethane, 3% propane, 1.5% butane, and 0.5% pentane. Each gas was given a DRE of 98% for simplicity. If the gas is assumed to be just methane (scenario M), then the calculated combustion efficiency is 98.6% as all of the $CO_2$ measured (produced by combustion of all the gases) is assumed to be due to combustion of just methane (with one carbon atom). With each additional gas (M + E = methane and ethane, M + E + Pr = methane + ethane + propane etc.), the calculated combustion efficiency trends towards the true combustion efficiency (98%). The second table shows an alternative scenario with a different (also fabricated) gas composition.

| Scenario | Assumed composition % | | | | | Total comp. | Calculated CE |
|---|---|---|---|---|---|---|---|
| | Methane | Ethane | Propane | Butane | Pentane | | |
| Actual composition | 85 | 10 | 3 | 1.5 | 0.5 | 100 | 0.9800 |
| M | 100 | 0 | 0 | 0 | 0 | 100 | 0.9860 |
| M + E | 89.5 | 10.5 | 0 | 0 | 0 | 100 | 0.9828 |
| M + E + Pr | 86.7 | 10.2 | 3.1 | 0.0 | 0 | 100 | 0.9814 |
| M + E + Pr + B | 85.4 | 10.1 | 3.0 | 1.5 | 0 | 100 | 0.9804 |
| M + E + Pr + B + Pe | 85 | 10 | 3 | 1.5 | 0.5 | 100 | 0.9800 |

Here the overestimation is more pronounced as methane makes up a relatively smaller amount of the overall gas composition.

| Scenario | Assumed composition % | | | | | Total comp. | Calculated CE / % |
|---|---|---|---|---|---|---|---|
| | Methane | Ethane | Propane | Butane | Pentane | | |
| Actual composition | 60 | 30 | 8 | 0.5 | 1.5 | 100 | 0.9800 |
| M | 100 | 0 | 0 | 0 | 0 | 100 | 0.9921 |
| M + E | 66.7 | 33.3 | 0 | 0 | 0 | 100 | 0.9843 |
| M + E + Pr | 61.2 | 30.6 | 8.2 | 0.0 | 0 | 100 | 0.9812 |
| M + E + Pr + B | 60.9 | 30.5 | 8.1 | 0.5 | 0 | 100 | 0.9810 |
| M + E + Pr + B + Pe | 60 | 30 | 8 | 0.5 | 1.5 | 100 | 0.9800 |

The following was added to the manuscript: "The extent of this overestimation depends on the exact composition of the fuel gas; the overestimation will be smaller the closer the proportion of $CH_4$ is to the assumed value of 100%."

Line 200: 'Eq. 3 will still overestimate the true combustion efficiency by some amount.' Similar to the previous comment, what is meant by 'some?'

See above response. This is difficult to quantify as it depends on the exact composition of the fuel gas.

Table 1: Is the assumption of 50% NO and 50% NO2 commonly used? I have seen some NOx studies use only NO2 when converting to mass, but I have not seen this 50/50 split before.

The reviewer is correct that most studies use the molar mass of $NO_2$ when converting $NO_x$ mole fraction into mass. However, this will undoubtedly end up with an overestimate of $NO_x$ mass. We have now used the average in-plume ratio of $NO:NO_2$ to calculate an average molar mass of $NO_x$ for each plume for use in mass conversions. The footer to Table 1 has been amended to:

"Uses an average molar mass for $NO_x$ calculated using the average in-plume ratio of $NO:NO_2$."

Line 358: Your NOx:CO2 ratios (0.003 ppm/ppm) are an order of magnitude larger than the values in Torres et al. (0.0002 ppb/ppb). What do you think explains this large difference?

The reason for the difference in measured $NO_x:CO_2$ ratios is unfortunately unknown. However, Torres et al. used manual test flares which may differ from real-world flares for many reasons. Firstly, the gas composition of the fuel gas was set as a mixture of natural gas and propane/propylene in $N_2$, which may not be representative of natural gas composition in the North Sea. Secondly, the gases were either steam- or air-assisted, and were flared at a set heating value,

which may be unrepresentative of flaring conditions in the North Sea. Thirdly, the flares were presumably consistently monitored and working, and were not in need of maintenance or malfunctioning, as may be the case for some flares in the North Sea. And finally, the environmental conditions in which the flares were tested (in Texas) were unlikely to be representative of the conditions in the North Sea. The following has been added to the manuscript:

"The reason for the order of magnitude difference between the $NO_x$:$CO_2$ ratios measured in this work and those reported by Torres et al. (2012c) is unknown, but is perhaps due to the specific flaring conditions measured in each case (Torres et al. measured emissions from manual test flares with targeted gas compositions and heating values, and not real-world flares operating in the North Sea)."

*Lines 384-389: Are the values from other works cited here specific to flaring or total emissions? If these other values are total emissions ratios, what does the comparisons mean?*

The reviewer is referring to comparisons of $C_2H_6$:$CH_4$ ratios measured in this work, against those reported by Wilde et al. (2021b) and Pühl et al. (*in prep.*) Their results did not specifically target flaring emissions, and therefore likely include some measurements of fugitive emissions (leaks) of natural gas from oil and gas infrastructure. In that sense, they are not direct comparisons, but are useful for context and perhaps could serve as an indication of the relative impacts of fugitive emissions vs. flared emissions on $C_2H_6$:$CH_4$ ratios. The following has been added to the text:

"It should be noted that the ratios measured by Wilde et al. (2021b) and Pühl et al. (in prep.) were not specifically attributed to flared emissions and were likely to be representative of total emissions from oil and gas infrastructure, including any vented emissions or fugitive natural gas leaks. Their ratios therefore cannot be compared directly against our own results but may serve as an indication of the relative impacts of flaring on $\Delta C_2H_6$:$\Delta CH_4$ ratios."

*Section 3.4: I am assuming that the inventory data shown in this section (and Figure 9) is for only flaring, but that is not explicitly stated anywhere.*

This is correct. The caption for Fig. 9 and the text has been updated. "Figure 9 shows the ECLIPSE $NO_x$:$CH_4$ emission ratio in the North Sea (for flared emissions) ...".

*Section 4: Do you have a sense of why the ECLIPSE inventory overestimates flaring methane emissions by such a large factor? The size of the discrepancy warrants a bit more discussion as to potential causes.*

One potential reason for the large overestimation is the coarse resolution of the ECLIPSE inventory. Looking at Figure D1, which shows $CH_4$ flaring emissions from both ECLIPSE and GFEI over the North Sea, the ECLIPSE inventory covers a much larger spatial area than GFEI. Whilst the slightly larger scale for GFEI emissions (on a per km basis) somewhat accounts for this, the cumulative emissions clearly do not match up.

Additionally, the ECLIPSE inventory for 2020 was created by projecting 2010-specific activity factors forwards into the future, based on legislative and Representative Concentration Pathways. If these pathways were not followed accurately over the decade succeeding 2010, then the 2020 inventory will be potentially inappropriate. Text concerning the ECLIPSE database descriptions has been amended – see comments below for changes made to the text in the Introduction and in Section 4.

[Figure]

*Lines 429-431: I do not see this global extrapolation number (7.6 Tg) in Plant et al., 2022. They do state their DRE_CH4 is ~91%, which is similar to the 92% used by the IEA to arrive at 8Tg of methane from global flaring.*

Apologies, this estimate (and the global extrapolation) was in an advance preview of the Plant et al. paper which we were communicated privately prior to submission and subsequent publication. Their calculated values still stand, however, but we have amended the text in our paper to read: "Using the effective $DRE_{CH_4}$ for onshore flaring (of 91.1%) measured by Plant et al. (2022) (which includes additional estimates of emissions from unlit flares), a total globally extrapolated emission of 7.6 Tg $CH_4$ from all onshore and offshore flaring can be estimated."

*Lines 479-482: In this concluding paragraph and previously in the results section, you discuss the skewed distribution of combustion efficiencies, but the median and mean values are close to expected. If I understand correctly, your emission estimate uses only this median value. So what does the skewness lead to, if anything?*

The skewed log-normal distribution has been observed both for total $CH_4$ emissions, and for flaring efficiencies. The distribution means that a few facilities (with very high emissions, or very low flaring efficiencies) are responsible for a large and imbalanced proportion of emissions - i.e., the distribution of emissions is unequal. As the frequency of these large emitters/low efficiency flares is quite low, it is possible that we underestimate their impact by 1) not capturing their true frequency in our sample set of measurements and 2) using a median value for flaring efficiency which is statistically closer to the high frequency of low emitters/high efficiency flares.

*Figure D1-2: It is interesting that no flaring emissions show up in the North Dakota, USA region. There are high flaring rates there. Does this suggest some other error in how ECLIPSE estimates flaring emissions?*

Yes, we agree with the reviewer here and thought this was interesting, and potentially indicative of problems with ECLIPSE. The ECLIPSE emissions dataset was created with the GAINS (Greenhouse gas – Air pollution Interactions and Synergies) model, which uses information about key sources of emissions from 172 country regions. GAINS relies on international and national statistics of activity data for energy usage (and other sectors). As ECLIPSE was built using 2010 activity data, it is possible that the inventory does not account for recent large-scale developments in the Bakken formation in North Dakota. ECLIPSE inventories after 2010 were created by projecting emissions forward based on legislation and Representative Concentration Pathways.

The description of the ECLIPSE inventory in the methods has been updated with: "ECLIPSE products used GAINS emissions data up until 2010, after which emissions were projected into the future using current legislation and Representative Concentration Pathways (Klimont et al., 2017)."

The discussion in Section 4 has been updated with: "The large difference in ECLIPSE estimated $CH_4$ flaring emissions could be a result of the inventory being a projected emission scenario for 2020, based on emissions representative of 2010 and legislation pathways (Klimont et al., 2017)." and "The nature of the ECLIPSE inventory estimates for 2020 (projected emissions based on 2010 emissions and legislation pathways) means that some major emission sources are missed. For example, no flaring emissions were ascribed to the Bakken formation region in the northern United States, despite recent (psot-2010) large-scale developments in shale gas there."

*Figure D3: Similar to the previous comment, there are no flaring emissions in the USA. That seems odd given it is one of the highest flaring nations.*

See above. It seems abnormal that no flaring emissions were prescribed to the USA, but this may be a result of the inventory being based on projections of emissions using 2010 emission activity data.

Author-made changes

Introduction

"Gas flaring is a practice widely used at hydrocarbon production sites to dispose of natural gas in situations where the gas is not captured for sale or used locally, and would otherwise be vented directly to atmosphere, or for reasons of safety. The World Bank defines three reasons for flaring: routine flaring, in which gas is flared during normal production operations; safety flaring, in which gas is flared to ensure safe operation; and non-routine flaring, which includes all flaring not incorporated by routine or safety flaring (World Bank, 2016)."

"Pohl et al. (1986) provided some of the first comprehensive measurements of flaring combustion efficiency, finding that flares operating with a stable flame achieved combustion efficiencies greater than 98%."

"Flares also differ widely in design and intended function, particularly between onshore and offshore, which will likely influence combustion efficiencies measured in different regions (Eman, 2015)."

Section 4

"Flaring in the UK North Sea reportedly fell by 23% in 2020 relative to 2019, but ~740 million cubic metres (7.4 × 108 m$^3$) of natural gas were still reported to have been flared (OGA, 2021)."

"In practice, flaring operations in the North Sea have some of the most stringent management systems due to a proactive regulatory regime."